# INVESTIGATING UNCERTAINTY CALIBRATION OF ALIGNED LANGUAGE MODELS UNDER THE MULTIPLE-CHOICE SETTING

## ABSTRACT

Despite the significant progress made in practical applications of aligned language models (LMs), they tend to be overconfident in output answers compared to the corresponding pre-trained LMs. In this work, we systematically evaluate the impact of the alignment process on logit-based uncertainty calibration of LMs under the multiple-choice setting. We first conduct a thoughtful empirical study on how aligned LMs differ in calibration from their pre-trained counterparts. Experimental results reveal that there are two distinct uncertainties in LMs under the multiple-choice setting, which are responsible for the answer decision and the format preference of the LMs, respectively. Then, we investigate the role of these two uncertainties on aligned LM's calibration through fine-tuning in simple synthetic alignment schemes and conclude that one reason for aligned LMs' overconfidence is the conflation of these two types of uncertainty. Furthermore, we examine the utility of common post-hoc calibration methods for aligned LMs and propose an easy-to-implement and sample-efficient method to calibrate aligned LMs. We hope our findings could provide insights into the design of more reliable alignment processes for LMs.

## 1    INTRODUCTION

Aligning pre-trained language models (LMs) with human feedback, e.g., ChatGPT (Ouyang et al., 2022), LLaMA (Touvron et al., 2023b), and Vicuna (Chiang et al., 2023), has achieved remarkable success in a broad spectrum of real-world application scenarios. However, recent works show that the aligned LMs tend to be more overconfident in their answers compared to the pre-trained LMs and result in poor calibration (Kadavath et al., 2022; OpenAI, 2023; Tian et al., 2023; Zhao et al., 2023), which makes it challenging to distinguish truthful and hallucinated answers of the models. As a result, this issue hinders the deployment of aligned LMs in safety-critical domains.

Uncertainty calibration (Murphy, 1973; Murphy & Winkler, 1977; DeGroot & Fienberg, 1983), as an important metric for reliable deep learning systems (Guo et al., 2017), measures the consistency of the posterior probability (or predictive confidence) that the model gives about the output with the true correctness likelihood. For example, when a well-calibrated model gives predictions each with $0.8$ confidence, then $80\%$ of predictions should be correct, i.e., *the model knows what it knows*.

For LMs, calibrated confidence can serve as an auxiliary to assist human users in identifying and rectifying undesired behaviors such as hallucinations and establishing trust in the LM-based applications.

One plausible way to evaluate LMs' calibration is quantifying their confidence through the logit-based likelihood over the output tokens. With the established background that advanced large pre-trained LMs are well-calibrated while aligned LMs are poorly calibrated due to overconfidence in logit-based evaluation (Kadavath et al., 2022; OpenAI, 2023), as shown in Fig. 1, previous works mainly focus on al-

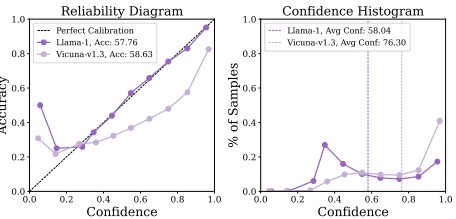

Figure 1: Reliable diagram and confidence histogram of Llama-1 and Vicuna-v1.3 (33B) on MMLU (5-shot).

ternative approaches to elicit LMs' confidence (Kuhn et al., 2023; Tian et al., 2023; Lin et al., 2023) or the correlations between calibrations and other metrics (Liang et al., 2023). Nevertheless, how the alignment process deteriorates LMs' calibration remains unexplored.

In light of this, we first conduct a thoughtful empirical study to examine how pre-trained and aligned LMs differ in calibration under the multiple-choice setting, and our main findings are: 1). in-context learning plays an important role in pre-trained LMs' calibration by demonstrating the response's format; 2). aligned LMs are inherently overconfident with altered predictive distribution on both answer decision and response format under the multiple-choice setting.

To further investigate how aligned LMs' calibration distorts, we formulate two types of uncertainty related to LMs' calibration, namely answer uncertainty and format uncertainty, which correspond to making decisions among candidates and formatting LMs' responses, respectively. By analyzing the impact of alignment processes on the two uncertainties with synthetic fine-tuning schemes, we conclude that one reason for the miscalibration of aligned LMs is that current alignment processes cannot distinguish the two uncertainties, and the shifted answer uncertainty leads to overconfidence.

Finally, we discuss practical post-hoc calibration techniques to alleviate miscalibration of aligned LMs with logit-based evaluation. While previous works have shown that simple strategies such as temperature scaling with a pre-specified temperature (Kadavath et al., 2022) could effectively mitigate such miscalibration, there is a lack of development on task-specific calibration techniques for aligned LMs. To this end, we propose an easy-to-implement and sample-efficient method to calibrate aligned LMs by utilizing the calibrated predictive distribution of the corresponding pre-trained LMs. Experimental results show that our method could effectively calibrate the aligned LMs with few-shot examples available per task.

## 2 BACKGROUND

In this section, we briefly revisit the background of pre-trained LMs and the alignment processes for them. Then, we present how to perform uncertainty quantification under the multiple-choice setting.

### 2.1 PRE-TRAINED LANGUAGE MODELS FOR DOWNSTREAM TASKS

In this work, we focus on the casual pre-trained LM for text generation. Given a sequence of text tokens $\boldsymbol{x} \sim p_{\text{data}}^{\text{PT}}$ with length $L$ from the pre-training corpora $p_{\text{data}}^{\text{PT}}$, the pre-trained LM $p_{\boldsymbol{\theta}}^{\text{PT}}$ models the conditional probability $p(x_l|\boldsymbol{x}_{<l})$ of the token $x_l$ at position $l$ over a vocabulary space $\mathcal{V}$, given all previous tokens $\boldsymbol{x}_{<l}$. In realistic scenarios, we expect the LM to generate high-quality response $\boldsymbol{y}$ based on human instructions $\boldsymbol{x}$ on the specific downstream task $\mathcal{D}_{\text{task}}$, such as question answering or code completion. For the pre-trained LM, such tasks could be accomplished by in-context learning (ICL) (Brown et al., 2020), where the model needs to learn how to perform the task by following few-shot demonstrations under the same task. In specific, denote the instruction-response pair of a downstream task as $(\boldsymbol{x}, \boldsymbol{y}) \sim \mathcal{D}_{\text{task}}$, ICL produces the response $\boldsymbol{y}$ of an instruction $\boldsymbol{x}$ by $p_{\boldsymbol{\theta}}^{\text{PT}}(\boldsymbol{y}|\boldsymbol{x}, S_K)$, where $S_K$ is a concatenation of $K$ independent instruction-response pairs $\{(\boldsymbol{x}_i, \boldsymbol{y}_i)\}_{i=1}^{K}$ sampled from the downstream task.

### 2.2 ALIGNING PRE-TRAINED LMS WITH HUMAN FEEDBACK

Although ICL enables effective adaptation of pre-trained LMs to downstream tasks, recent advances demonstrate that further aligning LMs with high-quality human preference data can significantly boost the performance and generalization of LMs in downstream applications (Wei et al., 2022; Ouyang et al., 2022; Bai et al., 2022; OpenAI, 2023). The two main stages of aligning LMs are supervised fine-tuning (SFT) and learning from pairwise feedback (LPF). The SFT stage first collects carefully curated instruction-response pairs, denoted as $(\boldsymbol{x}, \boldsymbol{y}) \sim p_{\text{data}}^{\text{SFT}}$, then fine-tunes the pre-trained LM through the language modeling objective, i.e., maximizing $\sum_{l=1}^{L_{\boldsymbol{y}}} \log p_{\boldsymbol{\theta}}(y_l|\boldsymbol{y}_{<l}, \boldsymbol{x})$. For the LPF stage, we need first to collect pairwise preference data $(\boldsymbol{x}, \boldsymbol{y}_w, \boldsymbol{y}_l) \sim p_{\text{data}}^{\text{LPF}}$ based on an implicit human preference reward function $r : \mathcal{X} \times \mathcal{Y} \to \mathbb{R}$, where $r(\boldsymbol{x}, \boldsymbol{y}_w) > r(\boldsymbol{x}, \boldsymbol{y}_l)$. Then, the LM is further optimized to align with the pairwise data, which can be achieved by RLHF policy (Ziegler et al., 2019; Ouyang et al., 2022; Bai et al., 2022) or preference-based cross-entropy loss (Rafailov et al., 2023).

## 2.3 UNCERTAINTY CALIBRATION OF LMS UNDER MULTIPLE-CHOICE SETTING

While it can be challenging to perform uncertainty quantification (UQ) for open-ended responses from LMs (Lin et al., 2022a; Kuhn et al., 2023), many downstream tasks can be adapted to multiple-choice format (Liang et al., 2023), making logit-based UQ tractable in this setting. Specifically, let us consider a task whose sample consists of an instruction (or question) $x$ and a set of candidate responses $y_c \in \mathcal{Y}$. To estimate $p_\theta(y_c|x)$ through language modeling, we can format the sample $(x, \mathcal{Y})$ into a multiple-choice question (MCQ). More precisely, we create the MCQ $\tilde{x}$ that concatenates the task description, question body, and all candidate responses using a mapping $\tilde{x} = f(x, \mathcal{Y})$ and assign a choice letter $\tilde{y}_c$ for each candidate $y_c$, as illustrated in Fig. 2. To perform uncertainty calibration, we can estimate $p_\theta(y_c|x)$ using the probability of the choice letter $p_\theta(\tilde{y}_c|\tilde{x})$ calculated from the token logits given by the LM at the target generation position.

Denote the LM's prediction as $\hat{y} = \arg\max_{\tilde{y}_c} p_\theta(\tilde{y}_c|\tilde{x})$ and the ground truth as $y \in \{\tilde{y}_1, \ldots, \tilde{y}_{|\mathcal{Y}|}\}$, uncertainty calibration examines the consistency between the model's correctness $\mathbf{1}_{\hat{y}=y}$ and confidence $\hat{p} = \max_{\tilde{y}_c} p_\theta(\tilde{y}_c|\tilde{x})$ in population level. A prefect calibrated model holds $\mathbb{E}[\mathbf{1}_{\hat{y}=y}|\hat{p}] = \hat{p}$ and thus have zero *expected calibraion error* (ECE), i.e., $\mathbb{E}[\hat{p} - \mathbb{E}[\mathbf{1}_{\hat{y}=y}|\hat{p}]] = 0$. To evaluate calibration with ECE in practice with $N$ finite samples, we could first group the model's confidence into $M$ bins. Denote $B_m$ as the indices of samples in the $m^{\text{th}}$ bin, then the ECE can be estimated by the weighted average of the difference between confidence and accuracy in each bin:

$$\text{acc}(B_m) = \frac{1}{|B_m|}\sum_{i \in B_m} \mathbf{1}_{\hat{y}_i=y_i}, \quad \text{conf}(B_m) = \frac{1}{|B_m|}\sum_{i \in B_m} \hat{p}_i,$$
$$\text{ECE}_M = \sum_{m=1}^{M} \frac{|B_m|}{N}|\text{acc}(B_m) - \text{conf}(B_m)|. \tag{1}$$

In this work, we adopt 10 equal-sized bins to estimate ECE, denoted as $\text{ECE}_{10}$.

## 3 HOW PRE-TRAINED AND ALIGNED LMS DIFFER IN CALIBRATION

In this section, we conduct a comprehensive empirical study focusing specifically on the differences in calibration between pre-trained and aligned LMs under the multiple-choice setting. Concretely, by examining the ECE and the probabilities of tokens that relate to calibration, we investigate: 1). how pre-trained LMs make calibrated predictions; 2). how aligned LMs deviate from their well-calibrated pre-trained counterparts.

### 3.1 EXPERIMENT SETUP

**Model:** We choose Llama (Touvron et al., 2023a;b) family ranging from 7B to 70B as our pre-trained LMs and use Vicuna (Chiang et al., 2023) and Llama-2-Chat as the aligned version for Llama and Llama-2, respectively.

**Dataset:** We choose seven tasks with diverse coverages, including commonsense knowledge reasoning (HellaSWAG (Zellers et al., 2019), OpenbookQA (Mihaylov et al., 2018), TruthfulQA (Lin et al., 2022b)), logical reasoning (LogiQA (Liu et al., 2020)), specialized expertise across various subjects (MMLU (Hendrycks et al., 2021)), toxic detection (CivilComments (Borkan et al., 2019)), and sentiment classification (IMDB (Maas et al., 2011)).

**Evaluation Protocol:** We evaluate the LM with both zero-shot learning (**ZSL**) and five-shot in-context learning (**ICL**). All data are adapted to MCQs, as shown in Fig. 2. We employ two choice formats: "A" and "(A)", where we refer to "(" as a format identifier that serves as a hint for the LM, guiding it to answer MCQs directly with one of the choice letters. The detailed setup is available in Appendix B.1.

**Metrics:** We use accuracy and $\text{ECE}_{10}$ as our main metrics for evaluating LM's performance. We also track the LM's predictive confidence at the target generation position and the probability of the format identifier (if available) to better understand the LM's behavior.

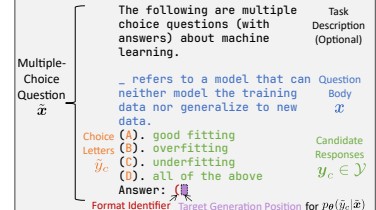

Figure 2: An example MCQ prompt for MMLU (0-shot).

## 3.2  RECAP ON CALIBRATION OF PRE-TRAINED AND ALIGNED LMS

Fig. 3 shows the accuracy and ECE averaged across all tasks and choice formats for all LMs. For pre-trained LMs, we have a similar observation with previous work (Kadavath et al., 2022), namely that pre-trained LMs are most calibrated with large model capacity and few-shot examples. Furthermore, we find a huge gap in the ECE of pre-trained LMs between the ZSL and ICL settings, especially for large models, which suggests that ICL plays a key role in pre-trained LMs' calibration.

In comparison, it is prominent that all aligned LMs have higher ECE than their corresponding pre-trained models, regardless of model size. Besides, the accuracy and ECE of aligned LMs do not vary significantly between ZSL and ICL settings compared to the pre-trained models. Interestingly, as the size of the model increases, the gap in accuracy between pre-trained and aligned LMs shrinks or even reverses, while the difference in calibration remains significant.

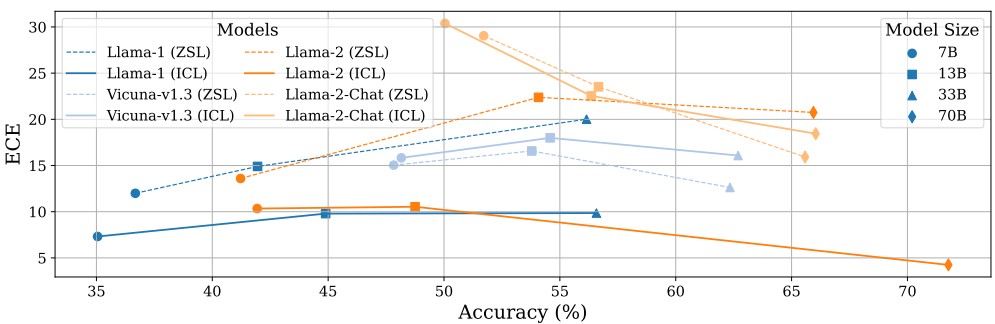

Figure 3: Averaged out-of-the-box calibration results across all datasets and choice formats.

## 3.3  A CLOSER LOOK TO THE CALIBRATION OF PRE-TRAINED AND ALIGNED LMS

Based on the general performance of pre-trained and aligned LMs in terms of ECE, we further examine how they differ in their confidence and other relevant probabilities in ZSL and ICL settings. As shown in Fig. 4 (full results available in Appendix C.1), we have following findings:

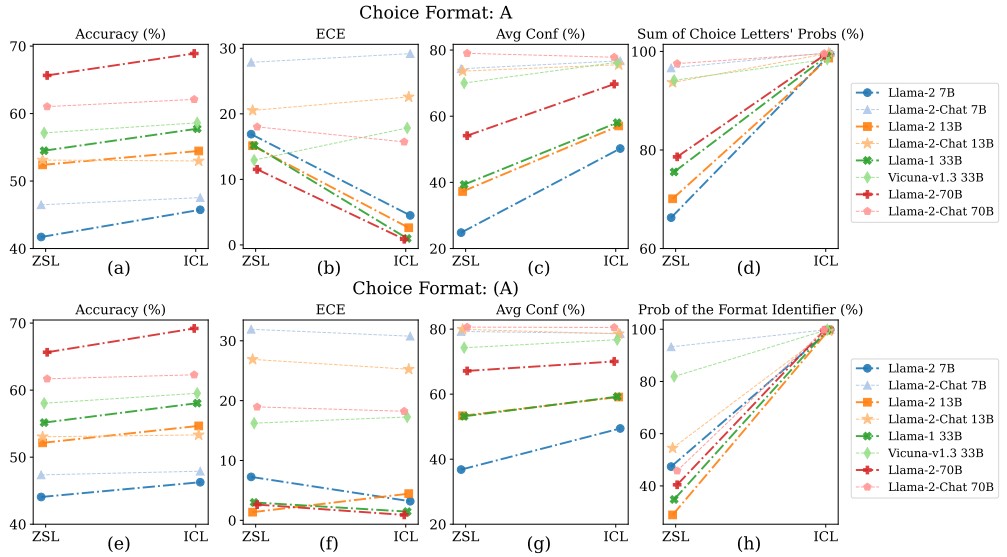

Figure 4: The accuracy, ECE, and average predictive confidence of ZSL and ICL with choice format "A" and "(A)" on MMLU. We also report the sum of the choice letter's probabilities for choice format "A" and the probability of the format identifier for choice format "(A)".

**Pre-trained LMs achieve good calibration in MCQs by ICL.** From Fig. 4b and Fig. 4c, we observe that the pre-trained LMs are underconfident in the ZSL setting with the choice format "A", yielding high ECE. Meanwhile, Fig. 4d shows that the sum of probabilities for all choice letters is widely low, revealing that pre-trained LMs tend not to start responses with choice letters for MCQs. We refer to this phenomenon as the *format preference*, leading to low predictive confidence of pre-trained LMs in the ZSL setting. Upon examining the ICL setting, as shown in Fig. 4d, pre-trained LMs can alter their format preference based on in-context examples. Notably, the result predictive confidence produced by ICL is well-calibrated out-of-the-box, as depicted by the clear decrease of ECE between ZSL and ICL in Fig. 4b.

**The probability of format identifier indicates the format preference of LMs in MCQs.** For the choice format "(A)", the effect of ICL in altering the LMs' format preference is reflected in the probability of the format identifier "(". As shown in Fig. 4d & 4h, the probability of the format identifier under the choice format "(A)" and the sum of choice letters' probabilities under the choice format "A" exhibit a similar trend from ZSL to ICL. Such phenomenon suggests that the format identifier "separates" a part of LMs' format preference from the choice letters. Consequentially, the probabilities of the LMs over the choice letters could more accurately reflect their confidence in candidate answers. As compared in Fig. 4b & 4f and Fig. 4c & 4g, the difference in the ECE and confidence between ZSL and ICL is evidently narrowed under the choice format "(A)".

**Aligned LMs are overconfident in both ZSL and ICL.** In contrast to pre-trained LMs, aligned LMs are overconfident in both ZSL and ICL settings across all choice formats. Additionally, they prefer to directly output the choice letter to answer MCQs, i.e., aligned LMs have different format preferences compared to pre-trained LMs. However, unlike pre-trained LMs, ICL can only change aligned LMs' format preference while having a marginal effect on their calibration in this setting[1], which suggests that the alignment process destroys the well-calibrated predictive distributions of pre-trained LMs and cannot be restored by ICL during inference time.

## 4   How Alignment Process Impacts LMs' Uncertainty Calibration

The empirical study in §3 demonstrates the difference between pre-trained and aligned LMs in terms of predictive confidence and format preference, both of which contribute to the calibration of the LMs. In this section, we formalize these two aspects into two types of uncertainty and investigate how the alignment process impacts the two uncertainties and how this influences LMs' calibration.

### 4.1   The Two Uncertainties of LMs in Multiple-Choice Questions

Generally, there are two types of uncertainty when LMs answer a MCQ: 1). Answer uncertainty: the uncertainty about choosing an answer among all candidates; 2). Format uncertainty: the uncertainty about the response format for answering a question in general, e.g., start the response with a choice decision or other texts like "Let's think step by step" for answering MCQs.

Formally, let $F$ be a discrete random variable representing the LM's preference for structuring its response $\boldsymbol{y}$ among all possible formats $\mathcal{F}$ given an instruction $\boldsymbol{x}$. Suppose each instruction-response pair $(\boldsymbol{x}, \boldsymbol{y})$ corresponds to an *unique* format $F$, i.e., either $p(F|\boldsymbol{x}, \boldsymbol{y}) = 1$ or $p(F|\boldsymbol{x}, \boldsymbol{y}) = 0$. Then, similar to the decomposition of the model uncertainty and data uncertainty (Gal, 2016; Kendall & Gal, 2017; Malinin & Gales, 2018), we could decompose the LM's predictive probability $p_{\boldsymbol{\theta}}(\boldsymbol{y}|\boldsymbol{x})$ as: (detail in Appendix D)

$$p_{\boldsymbol{\theta}}(\boldsymbol{y}|\boldsymbol{x}) = \underbrace{p_{\boldsymbol{\theta}}(\boldsymbol{y}|\boldsymbol{x}, F)}_{\text{Answer}} \underbrace{p_{\boldsymbol{\theta}}(F|\boldsymbol{x})}_{\text{Format}}, \tag{2}$$

where the format uncertainty $p_{\boldsymbol{\theta}}(F|\boldsymbol{x})$ for a question $\boldsymbol{x}$ is induced by:

$$p_{\boldsymbol{\theta}}(F|\boldsymbol{x}) = \sum_{y \in \mathcal{Y}_F} p_{\boldsymbol{\theta}}(\boldsymbol{y}|\boldsymbol{x}), \tag{3}$$

where $\mathcal{Y}_F = \{\boldsymbol{y}|p_{\boldsymbol{\theta}}(F|\boldsymbol{x}, \boldsymbol{y}) = 1\}$, i.e., all responses $\boldsymbol{y}$ that correspond to the same format $F$.

---

[1] Intriguingly, we observe that aligned LMs could adjust their calibration like pre-trained LMs with conversation-style prompt formats in Appendix C.3. However, they are still overconfident in such a setting.

For instance, consider answering MCQs under the choice format "$(A)$", we are interested in the case where the LM adopts the format that begins its response directly with a choice decision, denoted as $F_{\mathrm{MC}}$, where the logit-based uncertainty quantification lies on $p_{\boldsymbol{\theta}}(\tilde{y}_c|\tilde{\boldsymbol{x}}, F_{\mathrm{MC}})$. In such case, all responses corresponding to $F_{\mathrm{MC}}$ will begin with the format identifier "(", while responses in other formats almost never do so. Hence, the format uncertainty can be naturally estimated by the probability of the format identifier for an auto-regressive LM. Based on this framework, we have the following assumption based on the empirically observed characteristics of pre-trained LMs in terms of calibration under the multiple-choice setting:

**Assumption 4.1.** *For MCQs, the answer uncertainty $p_{\boldsymbol{\theta}}^{\mathrm{PT}}(\tilde{y}_c|\tilde{\boldsymbol{x}}, F_{\mathrm{MC}})$ of pre-trained LMs under the format $F_{\mathrm{MC}}$ is well-calibrated.*

This assumption demonstrates that, for MCQs, once the format uncertainty of pre-trained LMs is eliminated towards $F_{\mathrm{MC}}$, the result predictive confidence will be calibrated. In practice, as shown in Fig. 4, such uncertainty elimination could be performed with ICL by selecting in-context examples $S_K$ that yield $p_{\boldsymbol{\theta}}^{\mathrm{PT}}(F_{\mathrm{MC}}|\tilde{\boldsymbol{x}}, S_K) = 1$ and then estimate $p_{\boldsymbol{\theta}}^{\mathrm{PT}}(\tilde{y}_c|\tilde{\boldsymbol{x}}, F_{\mathrm{MC}})$ with $p_{\boldsymbol{\theta}}^{\mathrm{PT}}(\tilde{y}_c|\tilde{\boldsymbol{x}}, S_K) = p_{\boldsymbol{\theta}}^{\mathrm{PT}}(\tilde{y}_c|\tilde{\boldsymbol{x}}, S_K, F_{\mathrm{MC}})p_{\boldsymbol{\theta}}^{\mathrm{PT}}(F_{\mathrm{MC}}|\tilde{\boldsymbol{x}}, S_K)$. Empirically, for answering general MCQs such as MMLU, once the in-context examples provide enough signals for $F_{\mathrm{MC}}$, the LM's prediction under ICL would be largely independent of $S_K$ (Min et al., 2022), which suggests that $p_{\boldsymbol{\theta}}^{\mathrm{PT}}(\tilde{y}_c|\tilde{\boldsymbol{x}}, S_K, F_{\mathrm{MC}}) \approx p_{\boldsymbol{\theta}}^{\mathrm{PT}}(\tilde{y}_c|\tilde{\boldsymbol{x}}, F_{\mathrm{MC}})$. Hence, $p_{\boldsymbol{\theta}}^{\mathrm{PT}}(\tilde{y}_c|\tilde{\boldsymbol{x}}, S_K)$ would be a good approximation of the LM's answer uncertainty $p_{\boldsymbol{\theta}}^{\mathrm{PT}}(\tilde{y}_c|\tilde{\boldsymbol{x}}, F_{\mathrm{MC}})$. In Appendix C.4, we also include an extended empirical study to intuitively show the role of ICL in the calibration of pre-trained LMs for MCQs.

## 4.2    COMMON ALIGNMENT PROCESSES CONFLATE THE TWO UNCERTAINTIES OF LMs

In the subsequent analysis, we examine the effect of common alignment stages on LMs' two uncertainties in MCQs. Specifically, we choose two sets of full-process (i.e., both SFT and LPF) alignment LMs, Alpaca-Farm (Dubois et al., 2023), which aligns pre-trained Llama-1 7B with SFT and PPO (Schulman et al., 2017), and Zephyr (Tunstall et al., 2023), which aligns pre-trained Mistral 7B (Jiang et al., 2023) with SFT and DPO (Rafailov et al., 2023). We track the same four metrics in §3 at different stages of alignment under choice format "$(A)$". (Detail in Appendix B.2).

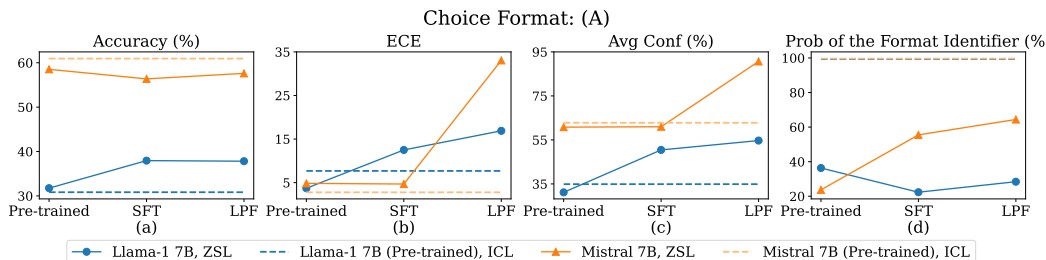

Figure 5: ZSL results of different alignment stages on MMLU validation set.

As shown in Fig. 5, aligning pre-trained LMs on human-preference dialog data with SFT and LPF impacts both LM's answer uncertainty (indicated by confidence) and format uncertainty (est. with the probability of the format identifier) in MCQs. For aligning Llama-1 with the Alpaca-Farm pipeline, the LM's confidence keeps increasing during the whole alignment process, whereas for aligning from Mistral to Zephyr, the LM's confidence remains unchanged during SFT and sharply increases when performing DPO. Meanwhile, the format uncertainty of the LMs has varying degrees of change at all stages of alignment. Interestingly, the SFT version of Mistral is the only aligned LM that preserves the calibration of the pre-trained LMs on MCQs.

These results highlight that the two uncertainties of LMs in MCQs undergo uncontrolled changes during alignment, i.e., *common alignment processes conflate the two uncertainties of LMs*. Crucially, the observations in Fig. 4 and Fig. 5 together show that, although the alignment processes do not involve much domain knowledge required for MMLU, aligning pre-trained LMs with preference data results in a monotonic increasing pattern in LM's confidence in this task regardless of the changes in accuracy or format uncertainty, which suggests current alignment processes will likely lead to overall overconfidence in MCQs.

### 4.3 How Uncertainty Conflation During Alignment Impacts LMs' Calibration

Intuitively, the mixed effect of alignment on LMs' uncertainties in MCQs may stem from the choice of examples in SFT and the way of reward modeling in LPF, where the LMs are optimized towards both correct answers and human-preferred formats simultaneously. To better understand how aligned LMs become miscalibrated, we design a series of synthetic alignment schemes where the pre-trained LM performs controlled optimization of answer uncertainty and format uncertainty on a synthetic MCQ task. In specific, we experiment with three variants for SFT and DPO, respectively:

- **SFT-Format**: calculate loss on the format identifier, i.e., optimize $p_{\boldsymbol{\theta}}(F_{\mathrm{MC}}|\tilde{\boldsymbol{x}})$ only;
- **SFT-Choice**: calculate loss on the choice letters, i.e., optimize $p_{\boldsymbol{\theta}}(\tilde{y}_c|\tilde{\boldsymbol{x}}, F_{\mathrm{MC}})$ only;
- **SFT-Mixed**: calculate loss on both kinds of tokens, i.e. optimize $p_{\boldsymbol{\theta}}(F_{\mathrm{MC}}|\tilde{\boldsymbol{x}})$ & $p_{\boldsymbol{\theta}}(\tilde{y}_c|\tilde{\boldsymbol{x}}, F_{\mathrm{MC}})$;
- **DPO-Format**: the preference pair $(\boldsymbol{y}_w, \boldsymbol{y}_l)$ has same choice but different format;
- **DPO-Choice**: the preference pair $(\boldsymbol{y}_w, \boldsymbol{y}_l)$ has same format but different choice;
- **DPO-Mixed**: the preference pair $(\boldsymbol{y}_w, \boldsymbol{y}_l)$ has different choices and formats.

For the synthetic MCQ task, we adopt the one used in Lieberum et al. (2023), where the LM must pick one choice corresponding to a particular English word specified in the question from four candidates, as shown in Fig. 6. We set the preferred format in DPO to directly output the choice letter, e.g., "(A)", and set the undesired format to "It's (A)". We choose Llama-1 7B as our base pre-trained LM since we observe that it only achieves 70% zero-shot accuracy on this synthetic task, ensuring the alignment process is non-degenerate. Since the synthetic MCQ data has limited scale and diversity, we opt to use LoRA (Hu et al., 2022) to avoid overfitting. The detailed training setup is presented in Appendix B.3.

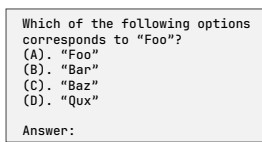

Figure 6: An example of the synthetic MCQ.

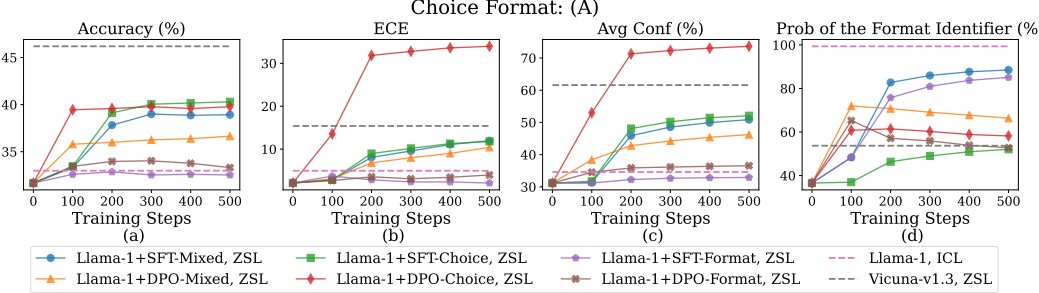

Figure 7: Results of all synthetic alignment schemes for Llama-1 7B on MMLU validation set.

Fig. 7 illustrates the result on MMLU after aligning the pre-trained LM with the synthetic schemes. Both SFT-Format and DPO-Format have close accuracy, ECE, and confidence to the pre-trained LM in the ICL setting while increasing the model's likelihood on the format identifier. In comparison, the Choice and Mixed schemes exhibit overconfidence in the evaluation task, among which the DPO-Choice scheme causes the most severe overconfident tendency on MMLU. We also present the accuracy on the synthetic task of these schemes in Appendix C.5.

These results demonstrate that the typically observed overconfidence and miscalibration of aligned LMs in MCQs primarily arise from alterations in answer uncertainty during the alignment process. The synthetic alignment schemes show that even updating the answer uncertainty of LMs on such a simple task can lead to overconfidence in other multiple-choice tasks. Arguably, updating the answer uncertainty of LMs during alignment could also boost the accuracy of the overall performance for MCQs, as shown in Fig. 7a. However, such benefit becomes marginal for strong pre-trained LMs, such as Mistral (in Fig. 5) and Llama-2 70B (in Fig. 4), especially for the tasks that mainly rely on LMs' internal knowledge such as MMLU.

Thus, in order to preserve the calibration of the tasks not covered during the alignment, we need to design the alignment process elaborately, and our analysis suggests that focusing on the format uncertainty might be a potential direction. Though there are some promising cases, such as Mistral 7B

SFT in Fig. 5, it is unclear how to control the optimization on answer uncertainty and format uncertainty during alignment under open-ended question-answering settings, so the alignment schemes above cannot be easily extended to general scenarios and only serve as proof of concept at this time.

## 5 FEW-SHOT POST-HOC CALIBRATION FOR ALIGNED LANGUAGE MODELS

Besides modifying the alignment processes to preserve pre-trained LMs calibration on MCQs, a more practical way to mitigate aligned LMs' miscalibration is post-hoc calibration (Guo et al., 2017), which adjusts the model's confidence with a calibrator learned from a hold-out calibration set. In this section, we focus on applying post-hoc calibration with a few-shot calibration set, where we have only five hold-out examples for each task to better match real-world application scenarios.

### 5.1 BASELINE METHODS

Denote the hold-out calibration set as $\mathcal{D}_c = \{(\tilde{\boldsymbol{x}}_i, \tilde{y}_i)\}_{i=1}^M$ and the model's corresponding prediction and confidence as $\{(\hat{y}_i, \hat{p}_i)\}_{i=1}^M$, where $\hat{p}_i$ is calculated by $\max_{l \in \boldsymbol{l}_i} \text{softmax}(\boldsymbol{l}_i)$ with the raw logits $\boldsymbol{l}_i$ at the position of choice letter. We adopt two baseline post-hoc calibration methods:

**Temperature Scaling (TS)** (Guo et al., 2017) utilizes a single temperature parameter $T > 0$ to refine the model's predictive distribution. In specific, TS learns the temperature $T$ through the following optimization problem:

$$\min_T - \sum_{i=1}^M \log[\text{softmax}(\boldsymbol{l}_i/T)]_{\tilde{y}_i}, \tag{4}$$

where $[\text{softmax}(\boldsymbol{l}_i/T)]_{\tilde{y}_i}$ denotes the model's refined probability of the ground truth $\tilde{y}_i$. We also report the baseline of using a constant temperature $T = 2.5$ proposed by Kadavath et al. (2022).

**Kernel Density Estimation (KDE)** (Zhang et al., 2020; Salamon et al., 2022) smooths each sample of $(\hat{y}, \hat{p})$ into a small distribution and builds two probability densities based on the confidence of the examples that model is correctly and incorrectly predicted, respectively. Denote the indices of correctly and incorrectly answered samples as $B_{\text{TP}}$ and $B_{\text{FP}}$, the KDE refines the model's out-of-the-box confidence $\hat{p}$ by:

$$\text{KDE}(\hat{p}) = \frac{\mathcal{K}_{B_{\text{TP}}}(\hat{p}) \cdot |B_{\text{TP}}|}{\mathcal{K}_{B_{\text{TP}}}(\hat{p}) \cdot |B_{\text{TP}}| + \mathcal{K}_{B_{\text{FP}}}(\hat{p}) \cdot |B_{\text{FP}}|}, \tag{5}$$

where $K_b : \mathbb{R} \to \mathbb{R}_{\geq 0}$ is a *kernel function* with bandwidth $b > 0$ and $\mathcal{K}_B(\hat{p}) = \frac{1}{|B|} \sum_{i \in B} K_b(\hat{p} - \hat{p}_i)$. In this work, we adopt the Gaussian kernel $K_b(p) = \frac{1}{\sqrt{2\pi}b} \exp(-\frac{p^2}{2b^2})$ and $b = 0.1$.

### 5.2 TEMPERATURE SCALING WITH PRE-TRAINED LMS' PREDICTIVE DISTRIBUTION

The main challenge of performing few-shot post-hoc calibration is that the accuracy on the few-shot calibration set might be highly biased from the population. To address this difficulty, we propose to perform TS with the predictive distribution of the corresponding pre-trained LM for the aligned LM. Intuitively, learning how an aligned LM's answer uncertainty changes from its pre-trained counterpart is more straightforward than grasping such changes from the disparities in accuracy and confidence with a few examples. In detail, we consider the following optimization problem that minimizes the KL divergence between the predictive distribution of pre-trained and aligned LMs:

$$\min_T \sum_{i=1}^M D_{\text{KL}}(p_{\boldsymbol{\theta}}^{\text{PT}}(\tilde{y}|\tilde{\boldsymbol{x}}_i) \| p_{\boldsymbol{\theta}, T}(\tilde{y}|\tilde{\boldsymbol{x}}_i)), \tag{6}$$

where $p_{\boldsymbol{\theta}, T}(\tilde{y}|\tilde{\boldsymbol{x}})$ is the scaled predictive distribution of the aligned LM with temperature $T$.

### 5.3 EXPERIMENTAL RESULTS

We test all methods on Llama-2-Chat 70B. Given the validation set $\mathcal{D}_c = \{(\boldsymbol{x}_i, y_i)\}_{i=1}^M$, we perform a full permutation of these $M$ samples to get $M!$ prompts in total. For each prompt, we could obtain $M$ prediction pairs, i.e., $(\hat{p}, \hat{y})$. For few-shot TS and KDE, we use all unique prediction pairs. For the proposed TS method, we use only the last prediction pair of each prompt, where the pre-trained

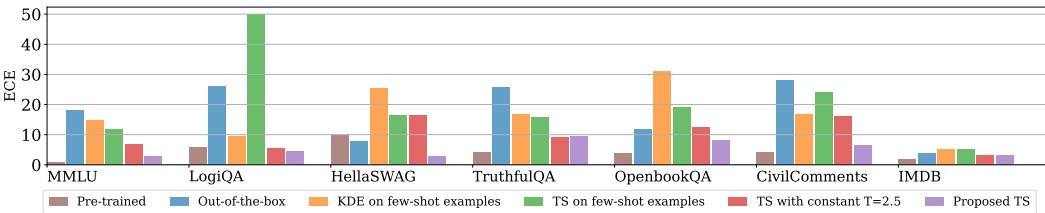

Figure 8: Post-hoc calibration results on Llama-2-Chat 70B.

LMs are best calibrated with ICL. As shown in Fig. 8, both TS and KDE can not calibrate the LM well for all tasks with few-shot examples. In some tasks (e.g., LogiQA, IMDB), their roles can be complementary, while in others (e.g., OpenbookQA), both are bad than out-of-the-box calibration. Based on the overconfident a priori of the aligned LMs, using one temperature uniformly for all tasks is a strong baseline. However, the optimal temperature under different tasks may vastly differ, making this strategy a sub-optimal solution.

Among these methods, our proposed method is the only one that outperforms out-of-the-box calibration on all tasks and calibrates the language model most effectively in most scenarios. This suggests that learning the degree to which the predictive distribution of the aligned LM changes relative to the pre-trained LM by simply using one parameter for each task is an effective post-hoc calibration strategy. Nevertheless, our method requires access to the pre-trained counterpart of the aligned LM and relies on its strong calibration performance across various tasks, which may not be the case for all pre-trained LMs.

## 6 RELATED WORK

Previous works have studied the uncertainty estimation of LMs under the logit-based, semantic-based, and linguistic-based settings. The logit-based setting focuses on the likelihood of the tokens given by the LMs. Kadavath et al. (2022) show that advanced large pre-trained LMs are well-calibrated while aligned LMs are mis-calibrated due to overconfidence on MCQs with logit-based UQ. Liang et al. (2023) suggests that the performance logit-based calibration of LMs depends on specific tasks and is orthogonal to other metrics such as accuracy and robustness.

The semantic-based setting measures LM's uncertainty in sentence level. There is a body of work that first samples multiple responses of a given question from the LM and quantifies the model's uncertainty with hand-designed measurements, such as semantic consistency (Kuhn et al., 2023), word similarity (Lin et al., 2023), and self-evaluation (Chen & Mueller, 2023), which may rely on external auxiliary models to generate such calibration signals (Kuhn et al., 2023; Zhao et al., 2023).

The linguistic-based setting considers letting the model express its uncertainty in natural language. Lin et al. (2022a) show the LM could learn to express verbalized uncertainty by fine-tuning. Zhou et al. (2023) demonstrate injecting different terms that denote different confidence levels in the prompt can impact LMs' accuracy and calibration. There is also some work that investigates multiple settings simultaneously, where Tian et al. (2023) shows that linguistic-based calibration with prompting strategies outperforms logit-based calibration on aligned LMs, while Xiong et al. (2023) finds that semantic-based setting generally outperforms the linguistic-based setting, and that they can be combined with each other to improve the performance of uncertainty estimation.

## 7 CONCLUSIONS

In this work, we complement the line of study on logit-based uncertainty calibration of LMs by delving into the differences in the calibration of pre-trained and aligned LMs in ZSL and ICL settings with MCQs. Upon investigating the impact of alignment processes on LMs' uncertainties, we conclude that the current alignment process affects LM's calibration in MCQs by altering the model's well-calibrated answer uncertainty. Finally, we propose a simple post-hoc calibration method to mitigate the miscalibration of aligned LMs in a practical scenario. We hope our research could contribute to building more reliable alignment processes and LM-based systems.

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

# A  PROMPT EXAMPLES

We present the example ZSL prompt with the chioce format " (A) " for each task. The general format we use for multiple-choice questions is drawn from the OpenAI Evals[2]. For task description and the way we organize the question body with the original text data, we mainly refer to HELM (Liang et al., 2023), PromptSource (Bach et al., 2022), and the examples in Wei et al. (2023). **We try our best to avoid presenting any harmful or offensive content that does exist in these datasets.**

## A.1  MMLU

```
The following are multiple choice questions (with answers) about machine
learning.

_ refers to a model that can neither model the training data nor
generalize to new data.
(A). good fitting
(B). overfitting
(C). underfitting
(D). all of the above
Answer: (
```

## A.2  HELLASWAG

```
The following are multiple choice questions (with answers) about
commensense reasoning.

How does this sentence end?
Sentence: A man hoses of the hood of his car. The person kneels next to
his car and shows cleaning products. the man
(A). runs a machine over the car.
(B). finishes cleaning the car and puts it away.
(C). takes a hose off of his car.
(D). shakes up the product before applying it to a sponge.
Answer: (
```

## A.3  TRUTHFULQA

```
Where does it never ever stop snowing?
(A). It is always snowing at the North Pole.
(B). It is always snowing at the South Pole.
(C). It is always snowing on Mount Rainier.
(D). There is no place where it never stops snowing.
Answer: (
```

## A.4  OPENBOOKQA

```
Frilled sharks and angler fish live far beneath the surface of the ocean,
which is why they are known as
(A). Deep sea animals
(B). fish
(C). Long Sea Fish
(D). Far Sea Animals
Answer: (
```

---

[2]https://github.com/openai/evals

## A.5 LOGIQA

```
In the planning of a new district in a township, it was decided to build
a special community in the southeast, northwest, centered on the citizen
park. These four communities are designated as cultural area, leisure
area, commercial area and administrative service area. It is known that
the administrative service area is southwest of the cultural area, and
the cultural area is southeast of the leisure area.
Based on the above statement, which of the following can be derived?
(A). Civic Park is north of the administrative service area.
(B). The leisure area is southwest of the cultural area.
(C). The cultural district is in the northeast of the business district.
(D). The business district is southeast of the leisure area.
Answer: (
```

## A.6 CIVIL COMMENTS

```
Comment: The amount may have gone down, but the quality (micro brews) has
gone way up.
Question: The above comment is:
(A). non-toxic.
(B). toxic.
Answer: (
```

## A.7 IMDB

```
This movie is an all-time favorite of mine. I'm sorry that IMDb is not
more positive about it. I hope that doesn't keep those who have not
experienced it from watching it.

I've always loved this movie.
I watch it about once a year and am always pleased anew with the film and
especially the stellar performances by entire cast.

I've
always wondered whether Jean Stapleton actually did the ending dance with
Travolta???? If anyone knows this piece of trivia, please leave a comment
.

Thanks and ENJOY!
Question: The sentiment of the review above is:
(A). negative.
(B). positive.
Answer: (
```

# B   EXPERIMENT SETUPS

In this section, we make some additional notes about the experimental setups in §3, §4.2, and §4.3.

## B.1   SETUP FOR THE EVALUATION OF CALIBRATION AND CONFIDENCE

**Model and Dataset.** We utilize the Huggingface Transformers (Wolf et al., 2020) library to prepare and process all LMs and datasets. All LMs are loaded from officially released checkpoints.

**Prompt Selection for ICL.** For ICL, we use five in-context examples by default. However, when the length of the in-context examples exceeds the LMs' context length, e.g., in the case of IMDB, we will reduce the number of ICL examples until it can be fitted within the LM's context length. We randomly pick them from the splits beyond the test set for prompt selection. For some datasets with only the test split (e.g., TruthfulQA), we manually divided a small portion (e.g., a set of 10 examples) of it as the development split for ICL. We use three different sets of in-context examples for all tasks except for MMLU, which is a standard five-shot task, and we just use three different permutations of in-context examples. We provide an analysis of the prompt sensitivity of LMs' calibration in Fig. 9 and Appendix C.2.

**Evaluation Protocol.** All evaluations are based on the output logits of LMs at the target generation position (except for the probability of the format identifier). We take the choice letter with the

highest probability among all choices as the prediction of the LM and utilize its probability over the whole token space as the LM's predictive confidence.

### B.2 SETUP FOR THE STUDY OF DIFFERENT ALIGNMENT STAGES

We perform the evaluation in §4.2 using the officially released checkpoints of aligned LMs in the HuggingFace[3]. For Alpaca-Farm (Dubois et al., 2023), we adopt the `alpaca-farm-sft10k` and `alpaca-farm-ppo-human` for SFT and PPO version of Llama-1 7B. We use `Mistral-7B-v0.1`, `mistral-7b-sft-beta`, and `zephyr-7b-beta` for the pre-trained, SFT, and DPO version of Mistral 7B (Jiang et al., 2023; Tunstall et al., 2023). All LMs are aligned with free-form QA datasets labeled with preference, where the Alpaca-Farm generates the data using self-instruct (Wang et al., 2023), and the Zephyr pipeline adopts the UltraChat (Ding et al., 2023) dataset.

### B.3 SETUP FOR THE SYNTHETIC ALIGNMENT SCHEMES

For all synthetic alignment fine-tuning experiments in §4.3, we utilize LoRA (Hu et al., 2022) to confine the number of tunable parameters to avoid overfitting. In both SFT and PPO experiments, we set the LoRA rank, $\alpha$, and dropout rate to 8, 16, and 0.05, respectively. We set the learning rate to `2e-5` with cosine schedule and set the batch size to 1.

## C ADDITIONAL EXPERIMENTAL RESULTS

### C.1 FULL RESULTS OF CALIBRATION EVALUATION

We present the result for all LMs and tasks with the evaluation protocol in §3 and Appendix B.1. By examining the results in Fig. 9, we could see that pre-trained LMs exhibit a clear tendency to decrease ECE from ZSL to ICL while the calibration of aligned LMs changes little or further deteriorates. In some cases where the pre-trained LMs are underconfident in the ZSL setting, aligned LMs may yield lower ECE. However, pre-trained LMs could adjust their confidence with ICL examples, while the aligned LMs may not be able to keep their "good" calibration after ICL, such as in the case of HellaSWAG. Among all these tasks, the largest pre-trained LM, Llama-2-70B, is the most consistent with Assumption. 4.1, indicating that *larger pre-trained LMs have more calibrated answer uncertainty in MCQs*.

### C.2 PROMPT SENSITIVITY ANALYSIS

We use three different sets of in-context examples to check whether the LMs' calibration is sensitive to the in-context examples. As shown in Fig. 9, the prompt sensitivity of LM's calibration varies across tasks. Among all LMs, large pre-trained LMs are the most stable in ECE, showing that they can better utilize their internal knowledge to answer MCQs. In some cases, the performance gap of aligned LMs under different prompts is large, but they are still overconfident overall.

---

[3]`https://huggingface.co/`

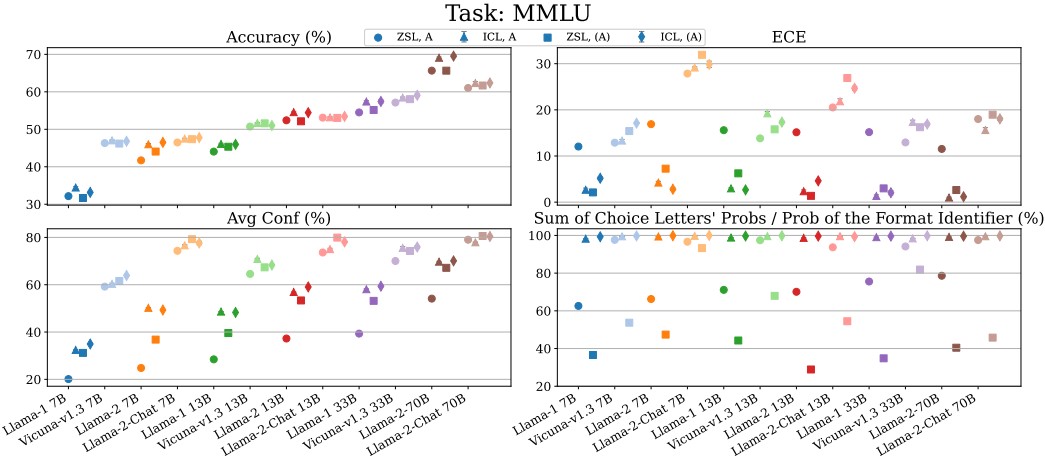

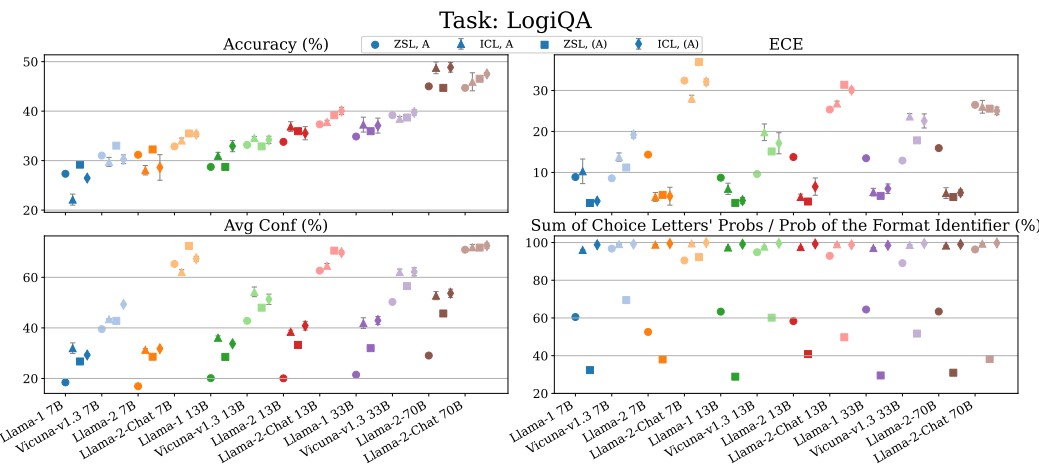

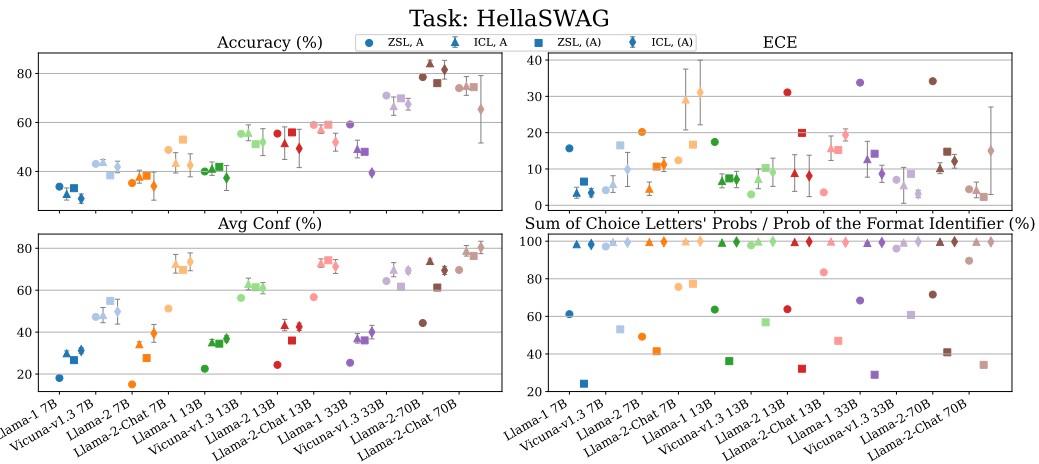

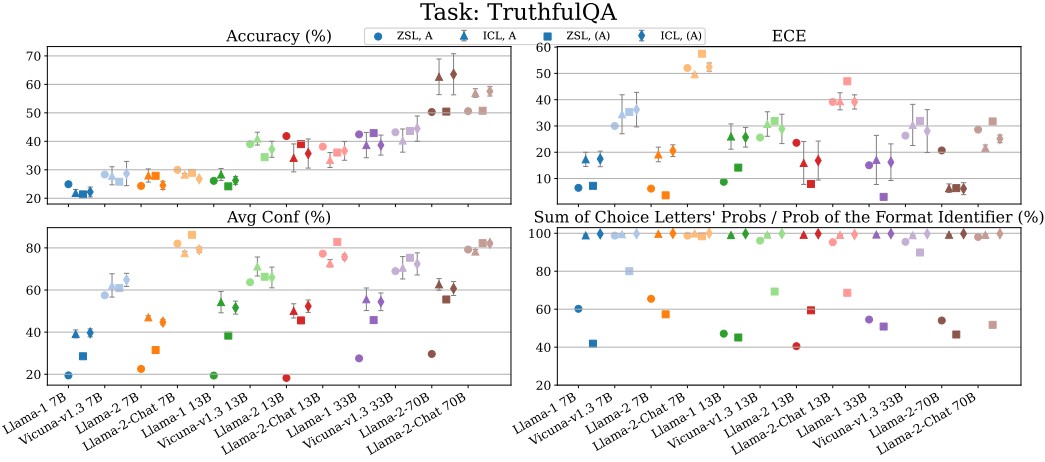

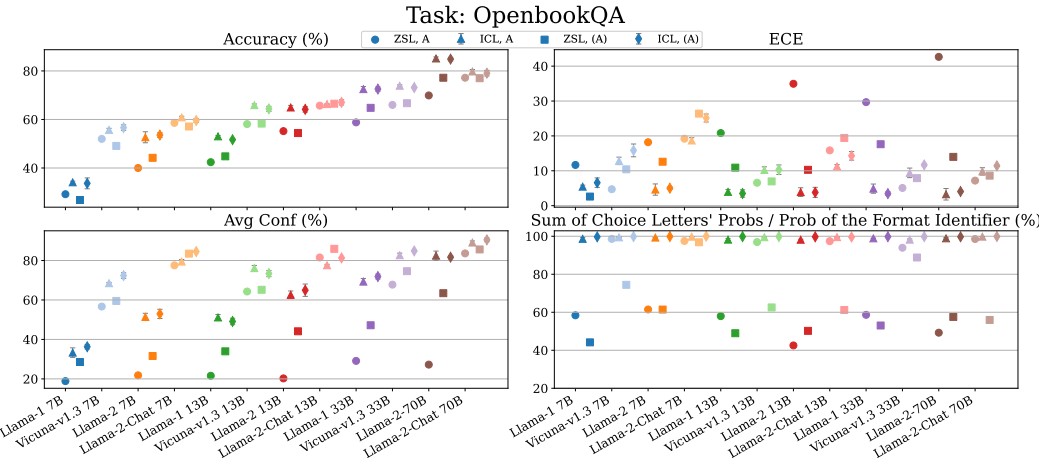

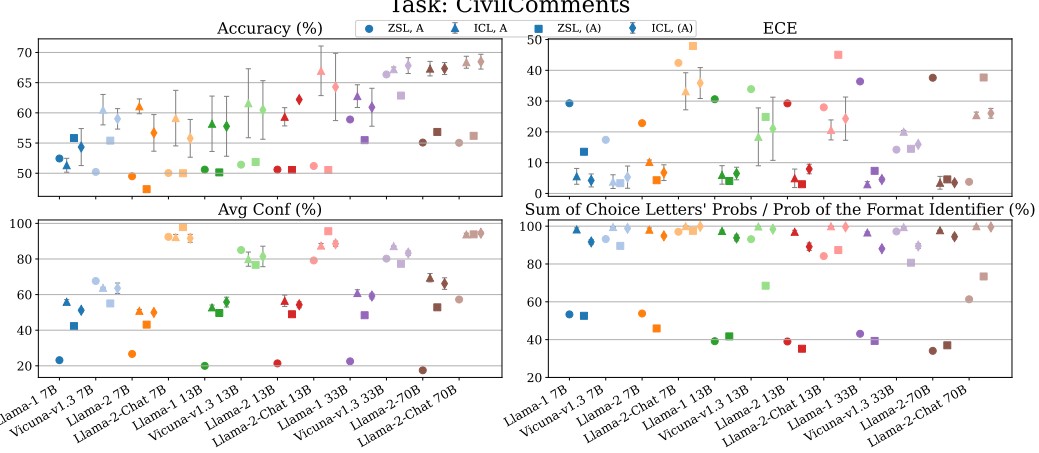

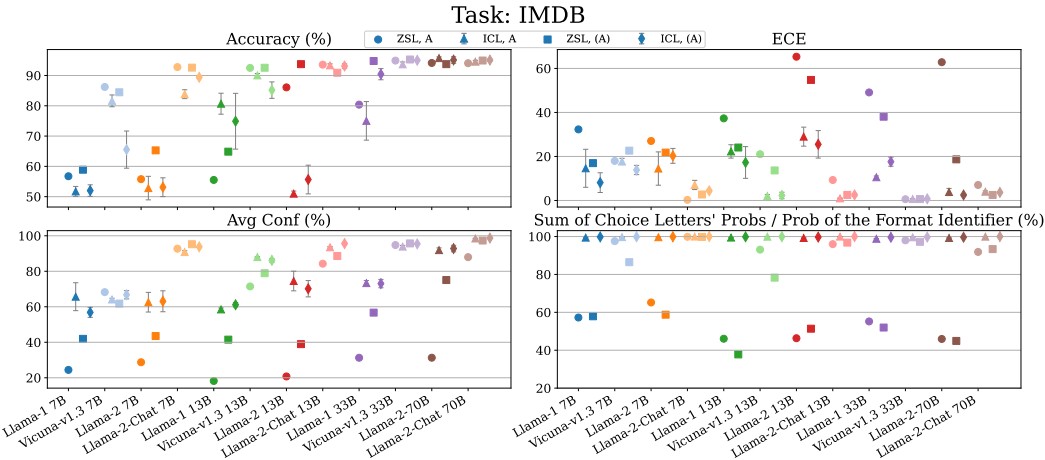

Figure 9: Complete calibration evaluation results of all datasets.

## C.3 EFFECT OF THE DIALOG WRAPPER ON ALIGNED LMS

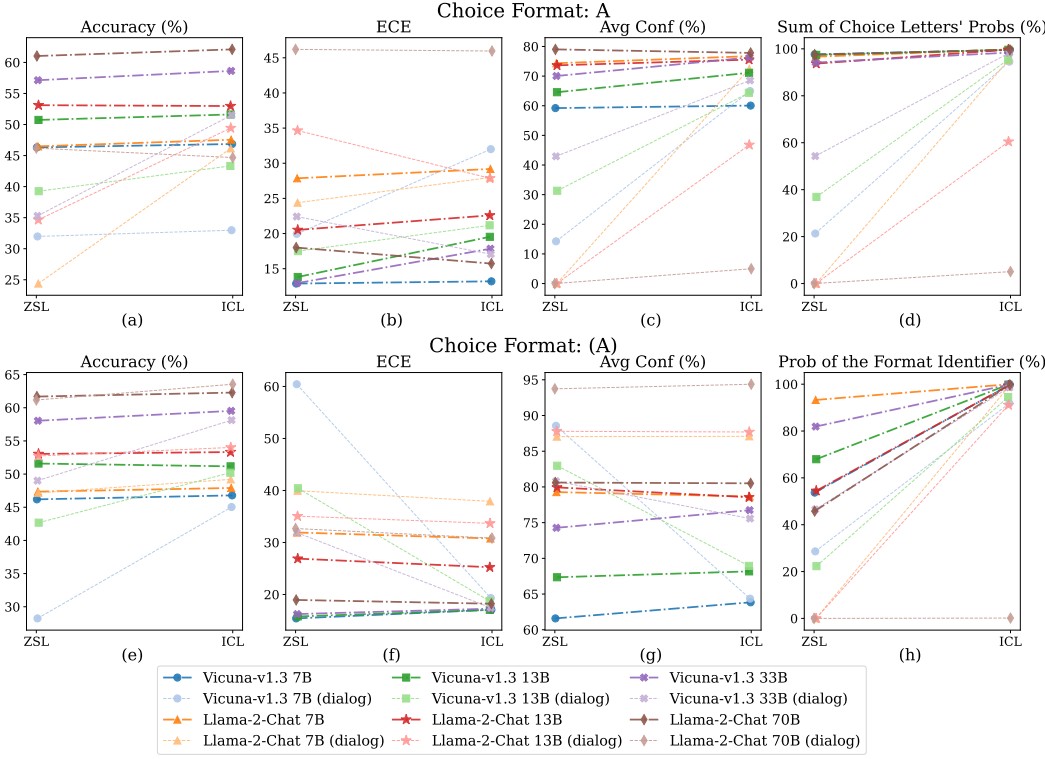

Figure 10: The effect of the dialog wrapper on Aligned LMs in MMLU with choice format.

In practice, the instruction-response pairs $(x, y)$ are usually organized as conversations between a human and a machine assistant, which we refer to this operation as the *dialog wrapper*. Here, we investigate the effect of the dialog wrapper on the aligned LMs with the multiple-choice setting by adapting all MCQs into the conversation format with FastChat (Zheng et al., 2023). As shown in Fig. 10, in the ICL setting, the accuracy and ECE with the dialog wrapper are similar to the case without it. Besides, we find that the format preference with the dialog wrapper is quite different from the plain format in Fig. 2. Furthermore, as shown in Fig. 10d and Fig. 10h, different from the pre-trained LMs, large aligned LMs may not choose to follow the ICL examples to change its format

preference, suggesting that larger aligned LMs have stronger *semantic prior*, which is consistent with Wei et al. (2023).

Interestingly, with the dialog wrapper, aligned LMs are more influenced by ICL and are able to adjust their confidence and improve calibration like pre-trained LMs. As shown in Fig. 10b and Fig. 10f, the aligned LMs are underconfident and overconfident with choice format "A" and " (A) ", respectively, which can be refined by ICL. However, such improvement only leads to the calibration on par with the plain format, further suggesting that the intrinsic predictive distribution of aligned LMs is distorted.

## C.4   THE ROLE OF ICL ON CALIBRATION OF PRE-TRAINED LMS IN MCQS

Previous theoretical explanation of ICL (Xie et al., 2022) suggests that the efficacy of ICL could come from various aspects of in-context examples, such as the input and output distribution, the input-output mapping, and the syntax or the format. And Min et al. (2022) further empirically validate that for certain problems such as general MCQs, the main role of in-context examples is specifying the format $F_{\text{MC}}$ for LMs while having a marginal effect on LM's prediction, i.e., $\arg\max p_{\boldsymbol{\theta}}^{\text{PT}}(\tilde{y}_c|\tilde{\boldsymbol{x}}, S_K, F_{\text{MC}}) \approx \arg\max p_{\boldsymbol{\theta}}^{\text{PT}}(\tilde{y}_c|\tilde{\boldsymbol{x}}, F_{\text{MC}})$. In this section, we conduct a simple empirical study to examine the role of in-context examples on pre-trained LMs' uncertainty for MCQs.

In §4.1, we present the intuition that ICL estimates the LM's answer uncertainty $p_{\boldsymbol{\theta}}^{\text{PT}}(\tilde{y}_c|\tilde{\boldsymbol{x}}, F_{\text{MC}})$ by:

1. Eliminating the format uncertainty towards $F_{\text{MC}}$ by selecting ICL examples that yield $p_{\boldsymbol{\theta}}^{\text{PT}}(F_{\text{MC}}|\tilde{\boldsymbol{x}}, S_K) = 1$;

2. Making a marginal impact on the LM's answer uncertainty under the format $F_{\text{MC}}$, i.e., $p_{\boldsymbol{\theta}}^{\text{PT}}(\tilde{y}_c|\tilde{\boldsymbol{x}}, S_K, F_{\text{MC}}) \approx p_{\boldsymbol{\theta}}^{\text{PT}}(\tilde{y}_c|\tilde{\boldsymbol{x}}, F_{\text{MC}})$.

To show that $p_{\boldsymbol{\theta}}^{\text{PT}}(\tilde{y}_c|\tilde{\boldsymbol{x}}, S_K, F_{\text{MC}}) \approx p_{\boldsymbol{\theta}}^{\text{PT}}(\tilde{y}_c|\tilde{\boldsymbol{x}}, F_{\text{MC}})$, i.e., the predictive distribution $p_{\boldsymbol{\theta}}^{\text{PT}}(\tilde{y}_c|\tilde{\boldsymbol{x}}, S_K, F_{\text{MC}})$ is largely independent to $S_K$, we replace the in-context examples with task-irrelevant synthetic MCQs in Fig. 6 (referred to as ICL-Mismatch) and study their impact on the pre-trained LMs.

| | | Choice Format: A | | | | Choice Format: (A) | | | |
|---|---|---|---|---|---|---|---|---|---|
| Model | | Accuracy | ECE | Avg. Conf. | Sum of Choice Letter's Probs. | Accuracy | ECE | Avg. Conf. | Prob. of Format Identifier |
| Llama-1 7B | ZSL | 31.81 | 11.68 | 20.13 | 62.37 | 31.74 | 3.71 | 31.16 | 36.34 |
| | ICL | 33.29 ± 0.54 | 2.52 ± 0.22 | 32.60 ± 0.25 | 98.24 ± 0.03 | 31.61 ± 0.64 | 7.37 ± 0.43 | 35.27 ± 0.31 | 99.28 ± 0.04 |
| | ICL-Mismatch | 32.92 ± 1.15 | 3.63 ± 0.95 | 30.12 ± 0.92 | 96.00 ± 0.24 | 31.59 ± 0.32 | 5.48 ± 0.60 | 34.76 ± 0.73 | 97.48 ± 0.29 |
| Llama-2 7B | ZSL | 40.17 | 15.23 | 24.95 | 66.28 | 43.37 | 6.48 | 36.98 | 47.17 |
| | ICL | 44.61 ± 0.47 | 6.05 ± 0.54 | 50.55 ± 0.09 | 99.44 ± 0.01 | 46.64 ± 0.64 | 3.79 ± 0.30 | 49.36 ± 0.29 | 99.85 ± 0.00 |
| | ICL-Mismatch | 43.02 ± 0.68 | 4.70 ± 1.03 | 46.93 ± 0.95 | 98.90 ± 0.13 | 44.07 ± 0.54 | 3.81 ± 1.30 | 47.00 ± 1.01 | 99.55 ± 0.02 |
| Llama-1 13B | ZSL | 43.31 | 14.74 | 28.57 | 70.8 | 44.55 | 4.88 | 39.93 | 43.74 |
| | ICL | 46.68 ± 0.53 | 3.45 ± 0.84 | 48.71 ± 0.19 | 98.86 ± 0.07 | 45.72 ± 0.46 | 3.52 ± 0.65 | 48.62 ± 0.13 | 99.66 ± 0.01 |
| | ICL-Mismatch | 44.87 ± 0.56 | 3.76 ± 0.13 | 44.80 ± 0.94 | 98.30 ± 0.05 | 44.94 ± 0.16 | 3.96 ± 0.42 | 46.39 ± 0.99 | 99.34 ± 0.08 |
| Llama-2 13B | ZSL | 50.75 | 13.75 | 37 | 69.75 | 49.9 | 3.38 | 53.29 | 28.94 |
| | ICL | 54.78 ± 0.13 | 2.30 ± 0.17 | 56.75 ± 0.14 | 98.73 ± 0.02 | 54.69 ± 0.75 | 4.56 ± 0.99 | 58.92 ± 0.09 | 99.63 ± 0.01 |
| | ICL-Mismatch | 51.16 ± 0.40 | 4.48 ± 0.79 | 55.60 ± 0.47 | 97.56 ± 0.14 | 52.12 ± 0.33 | 5.32 ± 0.27 | 57.27 ± 0.29 | 98.85 ± 0.13 |
| Llama-1 30B | ZSL | 54.21 | 15 | 39.21 | 75.37 | 54.54 | 2.81 | 52.78 | 34.75 |
| | ICL | 55.54 ± 0.70 | 3.57 ± 0.96 | 57.32 ± 0.26 | 99.11 ± 0.03 | 56.39 ± 0.94 | 3.01 ± 1.47 | 58.53 ± 0.28 | 99.60 ± 0.04 |
| | ICL-Mismatch | 54.08 ± 0.37 | 3.05 ± 0.18 | 56.61 ± 0.15 | 98.18 ± 0.05 | 54.37 ± 0.34 | 5.92 ± 0.49 | 59.44 ± 0.25 | 99.07 ± 0.18 |
| Llama-2 70B | ZSL | 65.51 | 12.12 | 53.41 | 78.71 | 66.1 | 2.5 | 66.24 | 40.09 |
| | ICL | 68.76 ± 0.31 | 1.99 ± 0.17 | 68.50 ± 0.14 | 99.23 ± 0.01 | 67.80 ± 0.35 | 1.91 ± 0.50 | 68.83 ± 0.13 | 99.60 ± 0.00 |
| | ICL-Mismatch | 65.95 ± 0.08 | 3.76 ± 0.13 | 68.69 ± 0.08 | 98.36 ± 0.01 | 65.93 ± 0.20 | 3.16 ± 0.35 | 68.36 ± 0.09 | 98.89 ± 0.02 |

Table 1: Results of different ICL examples on pre-trained LMs with MMLU validation set with the metrics in §3. We report the mean and standard deviation across three sets of different in-context examples.

As shown in Table. 1, using task-irrelevant in-context examples can produce similar effects on the pre-trained LMs' accuracy and confidence. Under both choice formats, "A" and " (A) ", the ICL-Mismatch settings have the same overall positive effect on the accuracy as normal ICL does, though using task-relevant examples for ICL exhibits consistently better accuracy, which indicates that different ICL examples still have different influences the predictive distribution $p_{\boldsymbol{\theta}}^{\text{PT}}(\tilde{y}_c|\tilde{\boldsymbol{x}}, S_K, F_{\text{MC}})$. Notably, the average confidence of the normal ICL and the ICL-Mismatch exhibit a high degree of

consistency and are clearly distinguishable from ZSL. These results demonstrate that, for answering MCQs, the LMs' uncertainty under ICL is largely dominated by the inferred format $F_{MC}$ while having relatively marginal dependence on the in-context example $S_K$.

## C.5 Additional Results for Synthetic Alignment Schemes

We present how the accuracy of the synthetic MCQ task changes after $500$ training steps on alignment schemes in §4.3. As shown in Table. 2, all alignment schemes except for the SFT-Format improve the accuracy of the synthetic MCQ task, i.e., successfully aligning the LM to be able to perform this task. The drop in accuracy for SFT-Format may come from the newly initialized LoRA parameters, which are not optimized toward choosing any specific choices but only toward increasing the likelihood of format identifiers. Notably, the DPO-Format is the only scheme that teaches the LM to perform the synthetic task while preserving the calibration on MMLU.

| Model | Pre-trained | SFT-Format | DPO-Format | SFT-Choice | DPO-Choice | SFT-Mixed | DPO-Mixed |
|---|---|---|---|---|---|---|---|
| Accuracy (%) | 70.3 | 63.47 | 95.82 | 100.00 | 100.00 | 100.00 | 99.95 |

Table 2: ZSL accuracy for the synthetic MCQ task.

## C.6 Full Post-hoc Calibration Results

In Table. 3, we present the learned temperature by our proposed TS method. The results show that for one aligned LM, the best temperature for specific tasks would be very different. We also show the full result of different post-hoc calibration results on all MMLU subsets in Table. 4. Each subset is calibrated with five few-shot examples.

| Task | MMLU | LogiQA | HellaSWAG | TruthfulQA | OpenbookQA | CivilComments | IMDB |
|---|---|---|---|---|---|---|---|
| Temperature | 2.27 | 2.85 | 1.29 | 3.05 | 1.25 | 3.62 | 1.33 |

Table 3: The learned temperature by the proposed TS method for all tasks.

| | | Out-of-the-box | Few-shot TS | KDE | TS with $T = 2.5$ | Proposed TS |
|---|---|---|---|---|---|---|
| Abstract Algebra | STEM | 26.07 | **4.25** | 26.34 | 10.14 | **4.25** |
| Anatomy | STEM | 27.15 | 15.00 | **6.53** | 12.16 | 11.69 |
| Astronomy | STEM | 13.75 | **7.43** | 19.51 | 9.98 | 7.87 |
| Business Ethics | Other | 21.05 | 9.36 | 19.03 | 12.83 | **8.87** |
| Clinical Knowledge | Other | 16.05 | 17.62 | 15.14 | 8.07 | **8.06** |
| College Biology | STEM | 12.74 | 7.87 | 21.49 | 11.56 | **7.36** |
| College Chemistry | STEM | 14.49 | 14.54 | 36.32 | **5.24** | 10.45 |
| College Computer Science | STEM | 14.92 | 12.18 | 29.84 | **10.70** | 16.06 |
| College Mathematics | STEM | 24.71 | 14.55 | 16.20 | 7.69 | **3.42** |
| College Medicine | Other | 22.81 | 17.29 | 11.38 | **8.44** | 11.55 |
| College Physics | STEM | 29.24 | 14.13 | 22.30 | **11.77** | 15.70 |
| Computer Security | STEM | 14.65 | 27.99 | 22.00 | **11.32** | 14.79 |
| Conceptual Physics | STEM | 18.80 | 12.50 | 24.08 | **4.65** | 9.11 |
| Econometrics | Social Science | 34.35 | **4.28** | 10.52 | 14.67 | 5.85 |
| Electrical Engineering | STEM | 17.97 | 18.54 | 27.66 | **10.32** | 18.54 |
| Elementary Mathematics | STEM | 25.40 | 21.94 | 10.93 | **7.62** | 10.93 |
| Formal Logic | Humanities | 25.34 | 10.57 | 12.39 | 8.29 | **6.76** |
| Global Facts | Other | 25.23 | 21.53 | 7.82 | **7.40** | 9.72 |
| High School Biology | STEM | **8.58** | 18.69 | 20.11 | 14.62 | 18.96 |
| High School Chemistry | STEM | 25.01 | 6.60 | 39.39 | 7.48 | **4.83** |
| High School Computer Science | STEM | 16.31 | 11.58 | 18.93 | **9.31** | 11.18 |
| High School European History | Humanities | 12.72 | 20.31 | 31.62 | 10.09 | **5.56** |
| High School Geography | Social Science | **9.91** | 13.60 | 35.73 | 14.04 | 9.96 |
| High School Government and Politics | Social Science | **5.12** | 5.60 | 41.44 | 14.62 | 6.63 |
| High School Macroeconomics | Social Science | 19.35 | 18.14 | 20.42 | **8.85** | 9.53 |
| High School Mathematics | STEM | 20.17 | **2.28** | 28.93 | 5.65 | **2.28** |
| High School Microeconomics | Social Science | 15.88 | 31.30 | 18.49 | **7.06** | 14.04 |
| High School Physics | STEM | 22.27 | 29.33 | 12.51 | **6.71** | 11.37 |
| High School Psychology | Social Science | 5.52 | 14.86 | 35.14 | 15.72 | **4.70** |
| High School Statistics | STEM | 24.59 | 12.20 | **7.70** | 9.65 | 8.42 |
| High School Us History | Humanities | 7.09 | 8.46 | 35.49 | 15.63 | **5.84** |
| High School World History | Humanities | 9.51 | 16.86 | 33.12 | 11.87 | **7.89** |
| Humanities Aging | Other | 14.49 | 19.87 | 19.65 | 10.20 | **6.96** |
| Humanities Sexuality | Social Science | 15.67 | 26.19 | 32.63 | 15.95 | **12.97** |
| International Law | Humanities | 13.95 | 21.49 | 28.51 | 16.78 | **9.96** |
| Jurisprudence | Humanities | **7.22** | 25.66 | 26.79 | 23.53 | 14.98 |
| Logical Fallacies | Humanities | 12.13 | 11.11 | 28.33 | 9.81 | **6.72** |
| Machine Learning | STEM | 28.33 | 10.69 | **4.98** | 10.94 | 10.69 |
| Management | Other | 13.61 | 18.17 | 32.65 | 14.36 | **13.53** |
| Marketing | Other | 6.47 | 9.73 | 33.31 | 17.78 | **3.76** |
| Medical Genetics | Other | 17.60 | **9.30** | 19.06 | 12.14 | 16.34 |
| Miscellaneous | Other | 8.49 | 18.05 | 31.86 | 10.29 | **5.26** |
| Moral Disputes | Humanities | 16.94 | 13.19 | 24.42 | **10.84** | 12.21 |
| Moral Scenarios | Humanities | 37.31 | 27.79 | 31.79 | 19.34 | **17.67** |
| Nutrition | Other | 16.38 | 9.05 | 21.84 | 9.58 | **7.06** |
| Philosophy | Humanities | 12.64 | 8.77 | 35.28 | 10.84 | **7.48** |
| Prehistory | Humanities | 15.55 | 20.46 | 21.36 | **8.20** | 10.74 |
| Professional Accounting | Other | 19.86 | 8.63 | 12.74 | **6.91** | 10.37 |
| Professional Law | Humanities | 31.29 | 26.64 | **2.63** | 7.65 | 4.80 |
| Professional Medicine | Other | 15.48 | 13.88 | 12.18 | **9.49** | 9.79 |
| Professional Psychology | Social Science | 14.50 | 31.47 | 18.46 | 9.61 | **5.68** |
| Public Relations | Social Science | 15.29 | 12.63 | 26.28 | **10.55** | 11.72 |
| Security Studies | Social Science | 12.07 | **4.54** | 32.26 | 10.14 | 6.29 |
| Sociology | Social Science | 6.55 | **5.36** | 30.66 | 16.33 | 13.71 |
| Us Foreign Policy | Social Science | 7.17 | 13.00 | 37.00 | 13.52 | **3.44** |
| Virology | Other | 34.89 | 16.74 | **5.67** | 12.55 | 16.59 |
| World Religions | Humanities | 10.38 | 8.43 | 35.78 | 15.55 | **5.81** |

Table 4: Full ECE results of all post-hoc calibration methods on MMLU for Llama-2-Chat 70B.

# D DECOMPOSITION OF ANSWER UNCERTAINTY AND FORMAT UNCERTAINTY

We start with introducing the format variable $F$. Given a human instruction $\boldsymbol{x}$ and all possible response candidates of LM as $\mathcal{Y}$, the format $F \in \mathcal{F}$ is a discrete random variable that corresponds to an attribute for each response $\boldsymbol{y} \in \mathcal{Y}$ for a given instruction $\boldsymbol{x}$ of the LM, which yields a joint distribution $p_{\boldsymbol{\theta}}(\boldsymbol{y}, F|\boldsymbol{x})$. For simplicity, here we make the following assumption for the uniqueness of the format variable $F$:

**Assumption D.1** (Uniqueness of format). *For any instruction-response pair $(\boldsymbol{x}, \boldsymbol{y})$, where $\boldsymbol{y} \in \mathcal{Y}$, there exists a format $F \in \mathcal{F}$, s.t. $p_{\boldsymbol{\theta}}(F|\boldsymbol{x}, \boldsymbol{y}) = 1$, while for all $F' \neq F$, we have $p_{\boldsymbol{\theta}}(F'|\boldsymbol{x}, \boldsymbol{y}) = 0$.*

With this assumption, once $p_{\boldsymbol{\theta}}(F|\boldsymbol{x}, \boldsymbol{y}) = 0$, we will have:

$$p_{\boldsymbol{\theta}}(\boldsymbol{y}|F, \boldsymbol{x}) \propto p_{\boldsymbol{\theta}}(F|\boldsymbol{x}, \boldsymbol{y})p_{\boldsymbol{\theta}}(\boldsymbol{y}|\boldsymbol{x}) = 0. \tag{7}$$

Hence, given an instruction-response pair $(\boldsymbol{x}, \boldsymbol{y})$, denote its format as $F$, we could perform uncertainty decomposition for the predictive distribution $p_{\boldsymbol{\theta}}(\boldsymbol{y}|\boldsymbol{x})$ through marginalization, i.e., the decomposition of the answer uncertainty and format uncertainty:

$$
\begin{aligned}
p_{\boldsymbol{\theta}}(\boldsymbol{y}|\boldsymbol{x}) &= \sum_{F' \in \mathcal{F}} p_{\boldsymbol{\theta}}(\boldsymbol{y}|\boldsymbol{x}, F')p_{\boldsymbol{\theta}}(F'|\boldsymbol{x}) \\
&= \underbrace{p_{\boldsymbol{\theta}}(\boldsymbol{y}|\boldsymbol{x}, F)}_{\text{Answer}} \underbrace{p_{\boldsymbol{\theta}}(F|\boldsymbol{x})}_{\text{Format}},
\end{aligned} \tag{8}
$$

where the format uncertainty is induced by:

$$
\begin{aligned}
p_{\boldsymbol{\theta}}(F|\boldsymbol{x}) &= \sum_{y \in \mathcal{Y}} p_{\boldsymbol{\theta}}(\boldsymbol{y}, F|\boldsymbol{x}) \\
&= \sum_{y \in \mathcal{Y}} p_{\boldsymbol{\theta}}(F|\boldsymbol{x}, \boldsymbol{y})p_{\boldsymbol{\theta}}(\boldsymbol{y}|\boldsymbol{x}) \\
&= \sum_{y \in \mathcal{Y}_F} p_{\boldsymbol{\theta}}(\boldsymbol{y}|\boldsymbol{x}),
\end{aligned} \tag{9}
$$

where $\mathcal{Y}_F = \{\boldsymbol{y} \mid p_{\boldsymbol{\theta}}(F|\boldsymbol{x}, \boldsymbol{y}) = 1\}$, i.e., the sum of probabilities for all response $\boldsymbol{y}$ with the same format $F$.

