# OpenReview forum: "Investigating Uncertainty Calibration of Aligned Language Models under the Multiple-Choice Setting"
_ICLR.cc/2024/Conference — Submitted to ICLR 2024_

### Official Review · Reviewer_2HUS · 2023-10-21

**Soundness:** 3 good
**Presentation:** 2 fair
**Contribution:** 2 fair
**Rating:** 6
**Confidence:** 3

**Summary:**

The authors investigate the calibration of "aligned" language models (finetuned and RLHF'd) on multiple choice tasks. They reproduce previous findings that aligned models are poorly calibrated and attribute this shortcoming to a conflation of two types of uncertainty: uncertainty about the format of the output and uncertainty about answers themselves. They show that targeted alignment intended to remove the former kind does not destroy calibration in the same way. Finally, they propose a temperature scaling technique for improving the calibration of aligned models post-hoc.

**Strengths:**

Calibrating aligned models is an important unsolved problem. While this paper is far from a solution, it contains some useful insights. It was interesting to see, for example, that almost all of the calibration error of unaligned models is simply a result of so-called "format error." Assuming it doesn't lower accuracy (see below), scaling temperature using the calibrated un-aligned model is also pretty neat.

**Weaknesses:**

The framing of sections 4.2 and 4.3 seems wrong to me. It is claimed that:

>Ideally, we would like the aligned LMs to generate responses with human-preferred formats and be equally calibrated as the pre-trained LMs, i.e., change the format uncertainty and preserve the answer uncertainty of pre-trained LMs, as ICL does.

While it's true that we'd want "aligned" models to be as or, better, more calibrated than the raw language models, that doesn't necessarily mean we'd want the *answer uncertainty* to remain the same. Essentially the same claim is repeated in Proposition 2:

>Proposition 2. Common alignment processes, such as SFT and LPF, alter both the answer uncertainty pθ(y|x, F) and the format uncertainty pθ(F|x) of LMs. Consequently, the changed answer uncertainty will cause the miscalibration of aligned LMs.

The whole point of "aligning" the models in this setting is to improve accuracy. It's difficult to imagine how to do that effectively while also preserving the approximate answer uncertainties of the original model. Enforcing this constraint would, it seems to me, produce unnecessarily brittle models with tiny margins of error. Of course, models with the same calibration error can have vastly different answer uncertainties (a model that outputs uniformly random answers is just as calibrated as a model that's confidently correct 100% of the time), and so there's no reason why high accuracy and good calibration couldn't be achieved while also tuning answer uncertainties. This issue is sidestepped in Figure 6 because accuracies remain roughly constant somehow (see the next paragraph), which makes me wonder why you'd even perform the intervention at all.

The experiments in section 3 are unclear. Figure 2 implies that the format identifier is provided as part of the prompt, and that the LLM simply has to output an answer choice. In Figure 4, though, there's a graph for the probability of outputting the format identifier, which seems to imply that the LLM is in fact outputting the format identifier itself. If so, what exactly are the plots on the second row of Figure 4 reporting? The average confidence across all generated tokens (the letter and also the two parentheses)? If so, how could e.g. the calibration error of models w/ ICL in 4f be the same as the corresponding figure in 4b? Since they're all correctly predicting the format specifier with 100% probability w/ ICL, it seems like the average calibration error should be significantly lower. Figure 6 has a similar issue. How could the accuracy of the blue + orange curves not increase along with the probability of outputting the format identifier? I could be misinterpreting what the authors mean here, but either way it would help to have a much clearer breakdown of what's being computed in these sections.

Bits and bobs:

- The notation in equation 6 isn't great. Would be better to incorporate the T.

**Questions:**

How does the proposed temperature scaling technique (Eq. 6) affect MC accuracy? Figure 7 just shows calibration error. I could imagine this intervention undoing some of the accuracy gains of the alignment process.

Re. figure 6, isn't it expected that finetuning on a binary classification task should wreck calibration on MMLU (effectively a 4-class classification task) since it de-weights choices C and D? Given that, I don't see how the rise in ECE can be attributed to increased confidence alone.

Also re. figure 6, it would be interesting to see Llama-1+Format, ICL.

---

> ### Author Response · Authors · 2023-11-19
> **Response to Reviewer 2HUS**
>
> Thank you for your effort and valuable comments and feedback! We would like to make some clarifications and answer your questions below:
>
> > W1: The framework of 4.2 and 4.3 is wrong. We want good calibration performance for alignment models but not necessarily the same answer uncertainty as the same pre-trained LMs.
>
> **A:** Thanks for pointing this out! We agree with your statement that changing the answer uncertainty does not necessarily cause overconfidence and miscalibration. Indeed, keeping the answer uncertainty of the pre-trained LMs is not desired during alignment since updating it could also boost the accuracy. Hence, the claim (such as "Ideally, ...") in our initial submission is inaccurate and inappropriate, and we removed them.
>
> Here, we want to make some clarifications about the somewhat convoluted phenomenon when aligning pre-trained LMs. The key point here is changing the answer uncertainty _in some human preference dialog data_ during alignment would _cause overconfidence in MCQs like MMLU_, which is usually beyond the scope of and not present in the dataset of alignment. This problem is similar to the in-distribution (ID) and out-of-distribution (OOD) settings, where optimizing the answer uncertainty on the ID dataset (in this case, the human preference data) would cause overconfident on the OOD dataset (in this case, the MCQs that is not presented in the alignment data). Tuning the answer uncertainty would result in good accuracy and calibration in the ID dataset, but the accuracy of the OOD dataset is basically unchanged, and the confidence is affected by the alignment process. This is especially true for larger LMs (e.g., Llama-2-70B in our experiments), where the alignment process does not improve the accuracy of MMLU but only increases confidence.
>
> In Section 4.2 of our revised version, we demonstrate that the common alignment processes update the LMs' answer \& format uncertainty simultaneously and cause overconfidence in MCQs, which is undesired since the alignment process does not involve much domain knowledge required for performing the MCQs in our evaluation. In Section 4.3, we add new synthetic schemes using pair-wise preference data with DPO and show that we could increase the accuracy of the synthetic task (illustrated in Table 1 in Appendix C.4) while preserving the calibration and changing format preference on MMLU.
>
> > W2: The experiments setup in Section 3 is unclear.
>
> **A:**
> Thank you for asking this. We add a reference in Section 3.1 to the detailed evaluation protocol in Appendix B.1. The accuracy, ECE, and confidence are calculated based on the target generation position for a choice letter (no matter which choice format, i.e., `A` or `(A)`, it is). So, these three metrics are computed based on **one** position that is supposed to be a choice letter.
>
> As for the probability of the format identifier, for an autoregressive LM, we could get all conditional probabilities $p(x_i|x_{<i})$ for all $x_i, i \in [2, L+1]$ within a single run of inference on the input, where $L$ is the length of the input text. Hence, we could get the probability of the format identifier without letting the LMs generate it by themselves.
>
> > W3: Equation 6 is not great.
>
> **A**: Thank you for your carefulness! We have fixed it.
>
> > Q1: How does the proposed TS technique affect MC accuracy?
>
> **A**: Temperature scaling preserves the accuracy since the argmax of the scaled distribution is still the same. The cost of TS is that it requires an additional calibration set for optimizing the temperature, which may not be available for all cases.
>
> > Q2: The binary-class QQP task will wreck the performance of the 4-class MMLU task.
>
> **A**: That's a great critical point. In response to that, we changed the dataset in Section 4.3 to a synthetic four-class MCQ dataset, where the LM is asked to pick the word that appeared in the question from four given options.
>
> > Q3: It would be interesting to see Llama-1 + Format, ICL.
>
> **A**: We present the results here (we use the setup in our revised version, which is conducted with Llama-1 7B):
>
>
>
> | Model     | Acc. | ECE  | Avg.Conf. | Prob. of `(` |
> | ----------- | ---- | ---- | --------- | -------------------------- |
> | Llama-1 ZSL           |   31.74   |  3.71    |     31.16      |          36.34                  |
> | Llama-1 ICL            |   30.82   |  7.69    |    34.89       |         99.33                   |
> | Llama-1 + Format, ZSL         |  32.52    |    2.10  |      32.92     |         85.11                   |
> | Llama-1 + Format, ICL | 31.03 | 7.23 |   36.16        |         99.91                   |
>
>
> The result shows that the performance of pre-trained Llama-1 and Llama-1 + Format has similar performance under ICL.

---

> > ### Comment · Reviewer_2HUS · 2023-11-21
> >
> > Thanks for the detailed response. I've raised my score from a 5 to a 6.

---

> > > ### Author Response · Authors · 2023-11-21
> > > **Glad to see our response helps**
> > >
> > > We are glad to have addressed your concerns and would like to thank you for your acknowledgment of our revised version. The improvements can't be made without your careful and constructive feedback.

---

### Official Review · Reviewer_rsG2 · 2023-10-30

**Soundness:** 2 fair
**Presentation:** 2 fair
**Contribution:** 2 fair
**Rating:** 5
**Confidence:** 4

**Summary:**

This paper studies how alignment training (supervised fine-tuning (SFT) and learning from pair-wise feedback (LPF)) affects the calibration ofv LLMs in the case of multiple-choice (MC). They decompose the uncertainty of the LLM's response into two components: (1) the answer uncertainty, which is the uncertainty about which option to choose, and (1) the format uncertainty, which is the uncertainty about how to format the answer. They show that alignment training makes the LLM overconfident by reducing both types of uncertainty. They propose a method to mitigate this overconfidence and demonstrate its effectiveness on a synthetic scheme. Last, they show that post-hoc calibration methods, including a proposed temperature scaling method, can improve the calibration of aligned LLMs.

**Strengths:**

- Calibration of LLMs is an important topic.
- Decomposing the uncertainty of LLMs into format uncertainty and answer uncertainty is novel.
- The background of the paper is properly introduced.

**Weaknesses:**

1. The paper contains many unclear sentences and claims.
    - Section 3.3: `Pre-trained LMs are calibrated in-context learners`. This sentence is confusing. Simply saying `Pre-trained LMs calibrate well in ICL setting` is enough.
    - Section 3.3:  `Format identifier decomposes format preference in MCQs`. This sentence and the whole paragraph are incomprehensible even when I read it again and again. I think this paragraph needs to be rewritten for better clarity.
    - Proposition 1 is unclear. I would expect a proposition to be something that can be derived from rigorous mathematical proof. The current content in Proposition 1 is only an observation based on the experiments. I also do not see why the first half of the sentence in Proposition 1 indicates (`i.e.`) indicates the second sentence. The following claim that $p_{\theta}^{PT}(y_c |x, F_{MC})\approx p_{\theta}^{PT}(y_c |x, F_{MC}, S_{K})$. I am not sure why the demonstrations do not affect the answer uncertainty about the LMs.
     - Proposition 2 has the same problem as Proposition 1. It is also unclear why changing the answer uncertainty will cause the miscalibration. SFT does not change the confidence but also changes the accuracy, so altering the answer uncertainty does not guarantee that the model will become miscalibrated. The following paragraph is also odd. `optimizing LMs with maximum likelihood in SFT encourages two uncertainties to be eliminated toward the SFT’s data distribution, and the impact on answer uncertainty will lead to aligned LMs’ overconfidence`.  The phrase `eliminated toward the SFT's data distribution` is unclear, and it is unclear why SFT leads to overconfidence in answer uncertainty. The current text seems to indicate that SFT fine-tuned models will be miscalibrated.

2. Many experiment settings are underspecified and hard to understand
    - How the ICL samples are selected is not specified. It is also unclear if the observations in this paper hold when varying the demonstrations in ICL.
    - The SFT and LPF datasets used in Figure 5 are not specified. It is thus unclear if the SFT dataset has a fixed or preferred format.
    - Figure 5 also does not explain why there are multiple points for LLama-2 between pre-trained and SFT models.

3. The figures are very difficult to read when printing on an A4 paper.
    - The dash lines in Figure 4 look very similar to non-dash lines.
    - The words (or symbols?) above the bars in Figure 7 are completely unreadable.

4. The method proposed in Section 4.3, only using the format for SFT, is highly limited to the MCQ setting and not straightforward to adapt to other settings. Moreover, this makes us unable to add new knowledge to the LLM during SFT. This is not very practical.

**Questions:**

Have you considered to instruct the LLMs to answer in the parentheses format to eliminate the format uncertainty instead of using ICL?

---

> ### Author Response · Authors · 2023-11-19
> **Response to Reviewer rsG2 (1/2)**
>
> Thank you for your devotion and careful review! We address the weaknesses and answer your question below:
>
> **Unclear sentences and claims**
>
> > W1: The sentence `Pre-trained LMs are calibrated in-context learners` is confusing.
>
> **A:** Sorry for the confusion. We want to highlight the fact that pre-trained LMs need ICL to be well-calibrated compared to ZSL. We revise the sentence to  `Pre-trained LMs achieve good calibration by ICL`.
>
> > W2: The sentence `Format identifier decomposes format preference in MCQs` and the corresponding paragraph is incomprehensible.
>
> **A:** We change the sentence to `The probability of format identifier indicates the format preference of LMs in MCQs.` and we rewrite the corresponding paragraph. The main point we want to convey is the difference between the choice format `A` and `(A)`, where the choice letter in the former case is responsible for representing both the answer and the format of MCQs, whereas in the latter case, the format preference is separate from the choice letters.
>
> > W3: Proposition 1 is unclear. I would expect a proposition to be something that can be derived from rigorous mathematical proof.
>
> **A:** Thank you for pointing it out. We admit that using propositions here is inappropriate, and we reframe this part to improve clarity. The revised version replaces the proposition with an assumption for pre-trained LMs that comes from the empirical observations that they have well-calibrated answer uncertainty $p_{\theta}^{\rm{PT}}(y_c|x, F_{\rm{MC}})$. And we intuitively and briefly discuss how ICL estimates $p_{\theta}^{\rm{PT}}(y_c|x, F_{\rm{MC}})$ with $p_{\theta}^{\rm{PT}}(y_c|x, S_K)$.
>
> To achieve this, ICL should (1) provide the signal for the format $F_{MC}$, i.e., $p_{\theta}^{\rm{PT}}(F_{\rm{MC}}|x, S_K) = 1$; (2) have minimal impact on the answer uncertainty of the test example, i.e., $p_{\theta}^{\rm{PT}}(y_c|x, F_{\rm{MC}}) \approx p_{\theta}^{\rm{PT}}(y_c|x, S_K, F_{\rm{MC}})$. A rigorous demonstration of (2) would require introducing a number of premises [1]. Here, we just utilize the intuition that when the LMs comprehend the MCQ task, their decision to the test example will not be influenced by the ICL examples since they are independent of each other.
>
> [1] [An Explanation of In-context Learning as Implicit Bayesian Inference](https://arxiv.org/abs/2111.02080) (Xie, Sang Michael, et al., ICLR 2022).
>
>
> > W4: Proposition 2 is unclear either.
>
> **A:** We admit that Proposition 2 could not provide a solid claim for explaining the source of the miscalibration of aligned LMs, and we remove that. The key point here is changing the answer uncertainty in some human preference dialog data during alignment would cause overconfidence in MCQs like MMLU, which is usually beyond the scope of and not present in the dataset of alignment. This is analogous to the problem of in-distribution (ID) and out-of-distribution (OOD), where optimizing the answer uncertainty on the ID dataset (in this case, the alignment dataset) will cause overconfidence on the OOD dataset (in this case, the general MCQs). Hence, for the ID dataset, the changed answer uncertainty and increased accuracy may result in a calibrated model, while for the OOD dataset, the accuracy remains unchanged, but the answer uncertainty is altered toward the direction of overconfidence. That's basically what happens when aligning the pre-trained LM and could not be demonstrated in the way in the original Proposition 2.
>
> In the revised version, we demonstrate this phenomenon based on the existing empirical results and support the claim that `altered answer uncertainty during alignment will cause overconfidence in MCQs` with our synthetic alignment schemes in Section 4.3.

---

> ### Author Response · Authors · 2023-11-19
> **Response to Reviewer rsG2 (2/2)**
>
> **Underspecified experiment settings**
>
> > W5: How the ICL samples are selected is not specified; it is also unclear if the observations in this paper hold when varying the demonstrations in ICL.
>
> **A:** Thank you for asking this. In the initial submission, we put the details about the selection of ICL samples and prompt sensitivity analysis in the Appendix without referring to them in the main text. In our revised version, we add the reference to these details in Section 3.1. For a short overview, we perform all evaluations with three sets of randomly selected in-context examples, and the conclusion is still valid. We also find that larger pre-trained LMs are more robust to the changes of ICL examples.
>
> > W6: The SFT and LPF datasets used in Figure 5 are not specified. It is thus unclear if the SFT dataset has a fixed or preferred format.
>
> **A**: We add a reference to the detailed information of these LMs in Appendix B.2 in our revised manuscript. The LMs in Fig.5 adopt common preference data similar to the aligned LMs in Section 3 (e.g., dialogs) for alignment. We use the official checkpoints to evaluate them on MCQs.
>
> > W7: Fig.5 does not explain why there are there are multiple points for Llama-2 between pre-trained and SFT models.
>
> **A**: They indicate different training steps during SFT. In our revised version, we replaced the Llama-2 with another model that employs DPO for the LPF stage for better demonstration.
>
> **Readability of Figures**
>
> > W8: Fig.4 and Fig.7 are unreadable
>
> **A:** Sorry for the inconvenience. We improved the readability of all figures in our manuscript.
>
> **About Section 4.3**
>
> > W9: The method proposed in Section 4.3 is not practical, highly limited to the MCQ setting, and not straightforward to adapt to other settings. Besides, it makes us unable to add new knowledge to the LLM during SFT.
>
> **A:** Thank you for asking this!
> We would like to clarify that the alignment schemes in Section 4.3 do not serve as novel practical solutions. Instead, by optimizing the answer and format uncertainty separately, their role is to help understand the source of miscalibration during alignment and to support that `altered answer uncertainty during alignment will cause overconfidence in MCQs`. However, our initial submission does not convey this point well, and we rewrite this section thoroughly.
>
> Besides, in our revised version, we add three pairwise preference learning schemes with DPO to make it closer to realistic settings and show that format learning with DPO could add new knowledge to the LLM (illustrated in Table 1 in Appendix C.4 due to space limitation) while preserving the calibration and changing format preference on MMLU.
>
> **Question**
>
> > Q1: Have you considered instructing the LLMs to answer in the parentheses format to eliminate the format uncertainty instead of using ICL?
>
> **A:** That's a great question! We tried this by adding the sentence like `Begin your response by selecting one option from [(A),(B),(C),(D)]` in the instruction, and we have the following ZSL results on MMLU (we mark the setting with such sentence as + Hint):
>
> For choice format A:
>
> | Model |  Sum of Prob. for [A, B, C, D]   |
> | ----- | ---- |
> | Llama-1 30B  | 75.54 |
> | Llama-1 30B + Hint  | 70.85 |
> | Vicuna-v1.3 33B  | 54.30 |
> | Vicuna-v1.3 33B + Hint| 90.14  |
>
>
> For choice format (A):
> | Model |  Prob. of `(`   |
> | ----- | ---- |
> | Llama-1 30B  | 34.81 |
> | Llama-1 30B + Hint  | 42.01 |
> | Vicuna-v1.3 33B  | 46.50 |
> | Vicuna-v1.3 33B + Hint  | 57.88 |
>
> While the hints could partly eliminate the format uncertainty of LMs, it is not as effective as in-context demonstrations for the LMs we evaluate.

---

> > ### Comment · Reviewer_rsG2 · 2023-11-21
> > **Re: response**
> >
> > Thank you for your clarification on the presentations.
> > However, I still cannot understand why we can assume the ICL examples do not affect the accuracy when the model understands the task.
> > Additionally, I cannot get the approximation stated in the paper ( $p_{\theta}^{PT}(y_c |x, F_{MC})\approx p_{\theta}^{PT}(y_c |x, F_{MC}, S_{K})$ ) based on the cited Bayesian ICL paper. I assume the authors are referring to Equation (7) in the Bayesian paper. However, the scenarios of the two papers are quite different, and as stated in the author's responses, it requires a lot of premises to state this formally. I cannot agree with the statements here if this approximation cannot be rigorously verified. I also disagree that ICL examples and the test sample are independent of each other. There has been a lot of evidence showing that ICL is very unstable due to the demonstrations.
> >
> > I appreciate the modifications in Section 4.3.
> >
> > I also appreciate the answers to my question.
> >
> > I raise the score to 5 (I am still slightly leaning against accepting the paper)

---

> ### Author Response · Authors · 2023-11-22
> **Response to Reviewer rsG2 (Round 2, 1/2)**
>
> Thank you so much for the discussion, which we believe is extremely helpful to further improve our work. We are also glad to see our revisions are recognized.
>
> We will try our best to address your remaining concerns and keep refining our work based on your valuable feedback. Here is our response:
>
> > Why can we assume the ICL examples do not affect the accuracy when the model understands the task?
>
> **A:** First, let's qualify the scope to ICL based on general MCQs, which is exactly the setting in our work. The fact that ICL examples have a marginal effect on the accuracy of MCQs has also been established by the previous empirical study [2]. The empirical study in [2] extensively demonstrates that randomizing the input-label mappings of ICL examples has a marginal impact on the accuracy of MCQs compared to the cases of using correct mappings, which, in our opinion, suggests that `When the LM understands the multiple-choice task, the LM's prediction under ICL would be largely independent of ICL examples`. Moreover, to better support the intuition that $p_{\theta}^{PT}(y_c |x, F_{MC})\approx p_{\theta}^{PT}(y_c |x, F_{MC}, S_{K})$, we add an additional empirical study in Appendix C.4 by using task-irrelevant MCQs as $S_K$ (refer to ICL-Mismatch) and compare the performance with the normal ICL. Here we present a brief overview of the results:
>
> Results on MMLU validation set under the choice format `(A)`:
>
> |    Model    |              |   Accuracy   |     ECE     |  Avg. Conf.  | Prob. of Format Identifier |
> |:-----------:|:------------:|:------------:|:-----------:|:------------:|:---------------------------:|
> |  Llama-2 7B |      ZSL     |     43.37    |     6.48    |     36.98    |            47.17            |
> |             |      ICL     | 46.64 ± 0.64 | 3.79 ± 0.30 | 49.36 ± 0.29 |         99.85 ± 0.00        |
> |             | ICL-Mismatch | 44.07 ± 0.54 | 3.81 ± 1.30 | 47.00 ± 1.01 |         99.55 ± 0.02        |
> | Llama-2 13B |      ZSL     |     49.9     |     3.38    |     53.29    |            28.94            |
> |             |      ICL     | 54.69 ± 0.75 | 4.56 ± 0.99 | 58.92 ± 0.09 |         99.63 ± 0.01        |
> |             | ICL-Mismatch | 52.12 ± 0.33 | 5.32 ± 0.27 | 57.27 ± 0.29 |         98.85 ± 0.13        |
>
> As shown in the result, the effects of ICL on the accuracy and confidence of pre-trained LMs are still valid when using task-irrelevant $S_K$ that corresponds to the correct format $F_{\rm{MC}}$. Although the task-relevant $S_K$ does provide more positive results in accuracy, it is not that significant compared to the totally task-irrelevant ones. Besides, the confidence in these two settings exhibits a strong consistency compared and is clearly distinguished from ZSL. We hope the additional results could help to convey our intuitive explanation better.
>
> > The approximation $p_{\theta}^{PT}(y_c |x, F_{MC})\approx p_{\theta}^{PT}(y_c |x, F_{MC}, S_{K})$ can not be established by solely citing [1].  I cannot agree with the statements here if this approximation cannot be rigorously verified.
>
> **A:** Thank you for pointing it out. We revised this by only referring to the established empirical evidence [2] in Section 4.1 and adding a clearer discussion about [1] and [2] in Appendix C.4.
>
>
> [1] [An Explanation of In-context Learning as Implicit Bayesian Inference](https://arxiv.org/abs/2111.02080) (Xie et al., ICLR 2022).
>
> [2] [Rethinking the Role of Demonstrations: What Makes In-Context Learning Work?](https://aclanthology.org/2022.emnlp-main.759) (Min et al., EMNLP 2022)

---

> ### Author Response · Authors · 2023-11-22
> **Response to Reviewer rsG2 (Round 2, 2/2)**
>
> >  I also disagree that ICL examples and the test sample are independent.
>
> **A:** We admit that it is not a rigorous way to support our intuition, and we avoid using such an expression in the paper.
>
> > There has been a lot of evidence showing that ICL is very unstable due to the demonstrations.
>
> **A:** Indeed, ICL does exhibit different biases in specific scenarios [3] or on some manually curated tasks where the LMs must rely on the in-context examples to be correct [4].
> However, when we qualify the setting to ICL for general, self-contained MCQs, we think there would be enough evidence to support the intuition that `The in-context examples have a marginal effect on LM's answer uncertainty for MCQs.`
>
>
> [3] [Calibrate Before Use: Improving Few-Shot Performance of Language Models](https://arxiv.org/abs/2102.09690) (Zhao et al., ICML 2021)
>
> [4] [Larger language models do in-context learning differently](https://arxiv.org/abs/2303.03846) (Wei et al, 2023)
>
> Overall, in order to make the intuition $p_{\theta}^{PT}(y_c |x, F_{MC})\approx p_{\theta}^{PT}(y_c |x, F_{MC}, S_{K})$ more sound, we revise our presentation by addressing the focus on ICL with general MCQs, adding the relevant empirical study as a reference, and using extra empirical evidence to support it, we also updated our manuscript accordingly. We hope our response could ease your concerns.
>
> Again, we appreciate your rigorous attitude and active engagement, which helps us keep polishing our work. We look forward to your further feedback and would be happy to answer any questions and take suggestions.

---

### Official Review · Reviewer_PsUf · 2023-11-01

**Soundness:** 2 fair
**Presentation:** 2 fair
**Contribution:** 2 fair
**Rating:** 5
**Confidence:** 4

**Summary:**

The paper studies the calibration of SoTA open-source LLMs in the context of multiple-choice QA. The paper focuses on the calibration of two variations of LLMs: standard and aligned and shows that aligned LLMs usually have worse calibration (as measured by ECE) as the alignment process increases overconfidence. Then, the paper shows that In-context learning seems to improve calibration with standard LMs by teaching the LM the desired answer format (which is not surprising). The paper shows that the two stages commonly employed in RLHF both contribute to the overconfidence issue of the LM.

Then, the paper studies approaches to calibrate aligned LMs and proposes a temperature scaling approach that relies on minimizing KL-divergence between the predictive distribution of the aligned LM and the same LM before alignment showing that it gives descent calibration performance and outperforms other approaches in terms of ECE.

**Strengths:**

* The paper studies an important issue.
* The study covers a wide variety of LLM families (LLama-1, LLama2, Vicuna) and sizes.
* The proposed temperature scaling approach is interesting and shows good calibration performance.

**Weaknesses:**

* The paper is mainly focusing on MCQ and while they argue that many tasks can be framed as MCQ tasks, many other tasks can not. Limiting the scope of the study to MCQ makes it unclear whether these findings will generalize to free-form generation, which I believe is more aligned with how LLMs are deployed in practice.
* I have to say that most of the findings in the paper are expected and some of them have already been established in the literature. For example, we know that alignment hurts calibration and we should expect ICL to mitigate issues with answer format uncertainty. I do not understand the motivation behind formalizing some of the paper findings as *propositions*. I do not think that adds anything to the paper. I would expect the paper to briefly discuss these expected findings and then move on to attacking the issue.
* The calibration-preserving training approach in section 4.3 is not novel; It is expected that training on the format only will likely reduce the overconfidence issue. In addition, there are two issues with the experimental setup there. First, the QQP dataset is not a pairwise preference ranking dataset, and therefore it's not entirely similar to the pairwise datasets used in realistic RLHF settings. Second,  the experiments were run with LoRA, again limiting the generalizability of these results to the true RLHF setting.
* The proposed temperature scaling approach is hard to apply in practice if we do not have access to the standard LM logits (the authors also point to this limitation) and is twice as computationally demanding as other approaches, while not being significantly better than simply using a constant temperature.

**Questions:**

* In section 5.3, you write that you use the last prediction pair of each prompt with your proposed TS method. Does that mean your approach effectively uses less data than the baselines? Have you tried using all pairs with your approach?
* Why did you pick the Quora-Question Pairs dataset? Why not a dataset designed for RLHF?




===== POST REBUTTAL ======

I thank the authors for their response. I will maintain my scores.

---

> ### Author Response · Authors · 2023-11-19
> **Response to Reviewer PsUF (1/2)**
>
> Thanks for your effort and valuable review! We address the weaknesses and clarify the questions below:
>
> > W1: The scope is limited to MCQ, and whether the findings will generalize to free-form generation is unclear.
>
> **A:** Indeed, our main concern is the calibration of LMs in MCQs, and we do not expect our findings to be naturally extended to free-form generation. We focus on understanding and mitigating the degradation of calibration from pre-trained to aligned LMs with logit-based UQ that originate from the evaluation of MCQs, where the LM's confidence is well-defined [1, 2]. For free-form generation, the pressing issue is how to quantify its confidence rationally, and logit-based UQ may not be a good solution. Hence, our work is somewhat orthogonal to the work related to UQ for free-form generation.
>
>
> > W2: Most of the findings in the paper are expected and have already been established in the literature. We should expect ICL to mitigate the miscalibration issues with answer format uncertainty.
>
> **A:** While some of our work's findings and results may align with expectations and intuitions, we believe that we still need to present them rigorously and comprehensively. Besides, the focus of our study is markedly different from existing work. Here is a brief overview of previous work related to logit-based UQ with MCQs in the context of pre-trained and aligned LMs:
>
> * [1]: Demonstrate the calibrated performance of the pre-trained LMs in MCQs and show that larger LMs with ICL are well-calibrated, with ECE and the reliability diagram as metrics. Show that the aligned LMs suffer from miscalibration and propose the TS with a constant temperature method.
>
> * [2]: Show that aligning GPT-4 with PPO degrades the ECE on a subset of MMLU with a single pair of reliability diagrams.
>
> * [3]: Present a comprehensive benchmark for all available LMs and compare different schemes (e.g., using the likelihood of sentence or choice letters) for logit-based calibration evaluation.
>
> It follows that even with the simple MCQ setting, there are still questions to be answered, such as **how _aligning LMs with human-preference data_ leads them to be _overconfident in MCQs_**, which is not quite straightforward to answer.
>
> Our study addresses this question by specifically pairing pre-trained LMs with their aligned counterparts for comparative evaluation. We report additional metrics previously unexplored in depth by other research, employ a framework for decomposing answer and format uncertainty in LMs, study different alignment stages, and design synthetic align schemes to investigate the different roles of the alignment process, which helps enhance the understanding of the source of miscalibration in aligned LMs.
>
> Besides, our work shows that ICL could only mitigate the miscalibration of pre-trained LMs caused by _underconfidence_ while having a marginal effect on aligned LMs' miscalibration due to _overconfidence_, which suggests the alignment process impairs the answer uncertainty of the pre-trained LMs. Again, we believe that this kind of difference between pre-trained and aligned LMs is not obvious and needs to be carefully discussed across model sizes, alignment algorithms, and tasks.
>
>
> [1] [Language Models (Mostly) Know What They Know](https://arxiv.org/abs/2207.05221) (Kadavath, Saurav, et al., 2023).
>
> [2] [GPT-4 Technical Report](https://arxiv.org/abs/2303.08774) (OpenAI, 2023).
>
> [3] [Holistic Evaluation of Language Models](https://arxiv.org/abs/2211.09110) (Liang, Percy, et al., TMLR 2023).
>
> > W3: Formalizing some of the paper findings as propositions is unnecessary and does not add anything to the paper. I would expect the paper to briefly discuss these expected findings and then move on to attacking the issue.
>
> **A:**
> Thank you for your feedback! We admit that the propositions are unnecessary and may cause confusion, and we accept your suggestion to remove Proposition 4.2 and re-organize the Proposition 4.1 part to an empirically established assumption with some intuitive discussions. We still preserve the formalization of the decomposition for a concrete demonstration of "answer" and "format" and enrich the experiment part with new results in Section 4, where we think careful discussion of the findings is necessary.

---

> > ### Author Response · Authors · 2023-11-19
> > **Response to Reviewer PsUF (2/2)**
> >
> > > W4: The calibration-preserving training approach is not novel, and it is expected that training on the format will likely reduce the overconfidence issue.
> >
> > **A:**
> > We would like to clarify that the alignment schemes do not serve as novel practical solutions. Instead, the role of them is to understand the source of miscalibration during alignment. However, our initial submission does not convey this point well, and we made substantial revisions to Section 4.3:
> >
> > (1) We modify this section's title to clarify its purpose.
> >
> > (2) We change the dataset to a synthetic MCQ task (see next answer).
> >
> > (3) We add new alignment schemes with DPO to better reflect the LPF case and support the idea that _the altered answer uncertainty during alignment causes overconfidence in aligned LMs_.
> >
> > > W5: There are two issues with the experiment setup in Section 4.3: (1) QQP is not a pairwise ranking dataset and is not entirely similar to the one used in realistic RLHF settings. (2) Running experiments with LoRA limits the generalizability of these results to the true RLHF setting.
> >
> > **A:** Thank you for pointing this out! We address the two points below:
> >
> > (1) The goal of the experiments in Section 4.3 is to understand the role of different components (i.e., optimizing answer and format uncertainty) of the alignment process through a simple and controllable setting. We hope a dataset that (a) could be formulated to MCQs to ease the decomposition of answer & format uncertainty; (b) should be simple enough that it does not require some complex background knowledge; \(c\) the pre-trained LMs could not handle it perfectly. Hence, the QQP, which is a simple paraphrase detection task, meets the above principles.
> >
> > Nevertheless, as pointed out by other reviewers, there is a gap between the binary-class QQP at training and, equivalently, four-class MMLU at evaluation so QQP is not a proper choice. Therefore, we choose to use a synthetic four-class MCQ dataset that meets the above principles. Besides, although we are not using a real RLHF dataset, we add three new alignment schemes based on preference learning using DPO, which is closer to the real RLHF setting compared to previous SFT schemes.
> >
> > (2) As mentioned in the paper, aligning a large LM with a single and small dataset will cause overfitting and hurt the general performance of other tasks, such as MMLU. Meanwhile, we do not expect these simple alignment schemes to be directly generalized to realistic settings, but rather that they will support our claims and provide some insights.
> > Therefore, we think optimizing the synthetic dataset with LoRA is enough to demonstrate the concept of this section, though there is indeed a gap between realistic alignment processes.
> >
> > > W6: The proposed temperature scaling method is (1) hard to apply in practice, (2) more computationally demanding, and (3) not significantly better than simply using a constant temperature.
> >
> > **A**: We have already admitted that (1) is the main limitation of our proposed method, as you mentioned. (2) is a tradeoff between better sample efficiency and computations. Although it's twice more computationally demanding, it could work well with few-shot examples, so we think the cost is acceptable. As for (3), we would like to address that _there is no principal way to determine this universal temperature_ and can only make a guess. So, there's a fair chance of making a bad guess to undermine the calibration further, as demonstrated in the case of HellaSWAG in Fig.8 of our revised manuscript.
> >
> > > Q1: Does the proposed TS method effectively use less data than the baselines? Have you tried using all pairs with your approach?
> >
> > **A:** That's a good question. Such a choice comes from the observation that pre-trained LMs are most calibrated after ICL, as presented in previous studies. In practice, using all pairs would have a similar overall performance to using the last pair. Since there is only one parameter (the temperature) to learn per task, the effect of bad predictive distribution of pre-trained LMs at first pairs is largely confined. However, we do observe that only using the first pair could cause degraded performance for some tasks, such as OpenbookQA, where the pre-trained LM is underconfident in ZSL, as shown in Fig.9 of Appendix C. One advantage of our method is that we do not need to access the ground truth label of the calibration set compared to the baselines.
> >
> > > Q2: Why choose the QQP dataset instead of a dataset for RLHF?
> >
> > **A:** Please refer to the answer of W5.

---

### Official Review · Reviewer_cVsf · 2023-11-02

**Soundness:** 3 good
**Presentation:** 4 excellent
**Contribution:** 2 fair
**Rating:** 6
**Confidence:** 3

**Summary:**

This paper investigates how the “alignment process” (instruction tuning, RLHF, etc…) affects the *uncertainty calibration* (does the models confidence on answers match of often it gets those answers right) of pre-trained language models (LMs). In particular, they focus on the tasks that can be formulated as multiple-choice tasks, as calibration can be computed more tractably for these ones.

The paper starts by measuring the calibration of open-source LLMs, across model families, multiple sizes per family, and pretrained vs aligned variants of each model. They confirm previous findings in the literature that pre-trained LMs are well-calibrated models, particularly when operating in a *in-context learning* (ICL) setting. Aligned LMs tend to be overconfident compared to their pre-trained counterparts (weather in a zero-shot or ICL setting), having considerable higher calibration error (ECE) compared to their pretrained counter parts.

The authors then identify two distinct uncertainties in LMs under multiple-choice settings: answer uncertainty for choosing among candidates, and format uncertainty for structuring responses. Current alignment processes conflate these two uncertainties, shifting the both format and answer uncertainty and leading to overconfidence in aligned LMs. In contrast zero-shot pretrained LLMs already have a good answer uncertainty and high format uncertainity, and ICL only changes/helps the latter. It also proposes a simple synthetic alignment scheme that purposely only modifies the format uncertainty, and this seems to help a zero-shot LLM get closer to the good calibration of LLMs on the ICL setting.

Finally the paper proposes a post-hoc calibration method for aligned LMs utilizing the calibrated predictive distribution of the pre-trained counterpart. Experiments show this can effectively calibrate aligned LMs with few-shot examples.

**Strengths:**

- I liked the investigation around the sources of mis-calibration and the distinction between the two types of uncertainty (answer vs format). Their synthetic alignment (altought limited) experiments also give evidence to their hypothesis that alignment messes with calibration mostly due to answer uncertainty.I think this analysis will be impactful in future research on calibration (at least for similar MCQ settings)
- Their analysis is quite-throughout, taking ICL, scaling and model family into account.

**Weaknesses:**

- They restrict their study to multiple-choice questions (MCQ). While understandable due tractability issues for open-ended generation, their identified problems and proposed solutions for calibration issues are only really applicable to MCQ settings and aren’t really generalizable to open-ended generation (what is “format uncertainty” in this case). This is especially problematic since the “alignment process” is normally applied through open-ended generation tasks. While I understand the difficulty in the open-ended analysis, some discussion on how the methods could generalize this setting (for example, the proposed temperature scaling should in principle work for open-ended generation aswell?) would help to ease this concerns
- Their proposed temperature-scaling with pretrained-LMs distribution requires access to the non-aligned pretrained LLM and a few examples *for every* new task that we might want to run our aligned model with. This might not always be feasible, particularly as closed-access LLMs proliferate and since of the goals of alignment is making them work well in the zero-shot scenario. Is there some way to generalize this such that we only really need to use the pretrained model once (for example, before the release of the aligned model)?

**Questions:**

n/a

---

> ### Author Response · Authors · 2023-11-19
> **Response to Reviewer cVsf**
>
> Thank you for your supportive comments and constructive reviews! We would like to address the weaknesses below:
>
> > W1: The study is restricted to MCQ, and the identified problems and proposed solutions for calibration issues are only applicable to MCQ settings. Some discussion on how the methods could generalize to open-ended generation would be helpful.
>
> **A:** Thank you for pointing this out. First, for the decomposition of answer \& format uncertainty and performing controlled optimization for them, we think this idea has the potential to be generalized in the open-ended generation. Although, as you mentioned, the quantification of format uncertainty, in this case, is not as straightforward as it is in MCQs. Nevertheless, we could still try to optimize the format preference only during general alignment processes.
>
> For example, in LPF, we could select the pair of $\boldsymbol{y}_w$ and $\boldsymbol{y}_l$ that share the same semantic meaning while structured with a different format. In our revised version, we design a new synthetic scheme following this idea by letting $\boldsymbol{y}_w$ = `(A)` and $\boldsymbol{y}_l$ = `It's (A)` and fine-tune the LM with DPO. The result aligned LM exhibits similar behavior to the original Format scheme using SFT, i.e., preserving the pre-trained LM's calibration on MMLU. Though such an alignment scheme still performs on MCQs, it's closer to the alignment objective on open-ended generation compared to only increasing the likelihood of the format identifier in the original Format scheme.
>
> Second, generalizing the proposed TS technique to open-ended generation scenario depends on whether the premises of the method are met. The key fact to making this approach work is that the pre-trained LMs are well-calibrated while the corresponding aligned LMs are not, which is well-established in MCQs. This would be true for short-form answers where our method could be naturally applied. However, to the best of our knowledge, for open-ended generation, the likelihood of a long answer may not be a proper indicator of LMs' confidence [1], and other forms of confidence quantification, such as linguistic confidence, do not show a clear superiority of pre-trained LMs over aligned LMs [2]. Hence, our proposed TS method may best fit the scenario similar to MCQs where logit-based UQ works well.
>
> [1] [Holistic Evaluation of Language Models](https://arxiv.org/abs/2211.09110) (Liang, Percy, et al., TMLR 2023) (The [ablation study of calibration in MCQs](https://crfm.stanford.edu/helm/latest/?group=ablation_multiple_choice)).
>
> [2] [Llamas Know What GPTs Don't Show: Surrogate Models for Confidence Estimation.](https://arxiv.org/abs/2311.08877) (Shrivastava, V., Liang, P., & Kumar, A, 2023) (Table 6 in Appendix A.6.).
>
>
> > W2: The proposed TS method requires access to the non-aligned LMs and a few examples for every new task that may not always be feasible. Is there some way to generalize this such that we only really need to use the pre-trained model once?
>
> **A:** We admit that our proposed TS method has the limitations you mentioned, and we also addressed them in the manuscript. In the context of post-hoc calibration, generalizing our task-wise approach to a model-wise one is not trivial since the degree of overconfidence is not quite the same on different tasks. Achieving this goal may need to include extra and potentially heavier procedures, and we would like to explore possibilities in future works.

---

> > ### Comment · Reviewer_cVsf · 2023-11-20
> > **Response to Rebuttal**
> >
> > I would like to thank the reviewers. Overall I found that the concerns brought up by the other reviewers are relevant and weren't sufficiently addressed, but still find the results in this work somewhat relevant for the community, so I'm keeping my score.

---

> ### Author Response · Authors · 2023-11-20
> **Thanks for the Reply!**
>
> Thank you again for your support and we sincerely appreciate your participation in the discussion phase.
> Also, if you don't mind, we respectfully hope you could be able to give us more feedback on your potential concerns, which remain unresolved by our rebuttal. We would be willing to address them.

---

### Author Response · Authors · 2023-11-19
**Common Response**

We thank all reviewers and ACs for your efforts and expertise. During this period, we have thoroughly revised the manuscript in response to all reviewer's valuable and helpful feedback. The major revision is marked with blue color.

In specific, the main change in the manuscript is improving the presentation of Section 4. With our basic experimental design and key results unchanged, we removed the inaccurate and inappropriate claims as well as refined and enriched the synthetic alignment experiment. We did our best to improve the presentation and clarity of the revised manuscript and respectfully request you check it out and re-evaluate our work.

## Summary of Revisions
* Remove the "summary of contribution" bullet list in the introduction due to space limit.
* In Section 3.1, we add a reference to Appendix B.1, which presents the detailed experimental setup (prompt selection, metric calculation).
* In Section 3.3, we add a reference to Appendix C.1, which contains the full evaluation results and the prompt sensitivity analysis.
* Revise the opening title of the first finding in Section 3.3.
* Rewrite the second finding related to the format identifier in Section 3.3 for better clarity.
* Substitute Proposition 1 with an assumption that comes from the empirical observation in Section 3. And rewrite the reasoning part in Section 4.1.
* Revise the title of Sections 4.2 and 4.3 to convey the main idea better.
* Remove Proposition 2 in Section 4.2 and the related vague and inappropriate expressions.
* Add the result of DPO-aligned LMs in Section 4.2 and rewrite Section 4.2 based on the empirical observation to highlight the claim that alignment processes conflate the answer and format uncertainty.
* Change the dataset used in the synthetic experiment in Section 4.3 from QQP to a four-class synthetic MCQ dataset and add three new synthetic alignment schemes based on DPO.

We also address the weakness and questions raised by reviewers correspondingly in detail and would like to clarify any further concerns. We are welcome to any further discussions and open to keep improving our work in all aspects.

---

### Meta-Review · Area_Chair_Eo7A · 2023-12-06

**Metareview:**

This work investigates the calibration of language models on Multiple-Choice-Questions (MCQs), and considers the effect of “alignment” on calibration. Consistent with prior works, they find that alignment often hurts calibration. Their main observations are: (1) There are two types of errors in MCQ generations: “format errors” (e.g. not producing a valid choice), and “true” answer errors. Evaluation of LMs should delineate these two types. (2) In-Context-Learning (providing examples of other properly-formatted questions from the relevant test set) helps significantly for pre-trained models (largely by fixing format-errors), but not for aligned models.

Reviewers acknowledged these two contributions, and liked the initial investigation in synthetic settings. Ultimately, this paper shows promise, but reviewers did not identify sufficiently strong contributions to allow acceptance at ICLR.
Notable weaknesses which prevented a higher score:
* The paper studies a restricted setup (MCQ), where calibration can easily be measured, but “alignment” is typically more relevant in more general settings.
* The observations on format-errors an ICL are interesting and worth pointing out, but are not especially insightful on their own. (E.g. several reviewers noted that this type of ICL is expected to improve format errors).
* There is not much investigation of *why* alignment leads to mis-calibration, and why it suppresses the benefit of ICL. The answer to this may depend a lot on the particular methods of pre-training and alignment, and these factors are not thoroughly studied in this work.
* The proposed re-calibration methods are not particularly novel nor practical.

I recommend rejection for ICLR. I encourage the authors to continue pursuing this direction to strengthen the results.

**Justification For Why Not Higher Score:**

Reviewers did not identify sufficiently strong contributions. Notable weaknesses are listed above; briefly, the setting is restricted, the observations are not especially insightful, and the mechanisms are not investigated.

**Justification For Why Not Lower Score:**

N/A

---

### Decision · Program_Chairs · 2024-01-16

Reject